# Beyond Turn Limits: Training Deep Search Agents with Dynamic Context Window

## Abstract

While recent advances in reasoning models have demonstrated cognitive behaviors through reinforcement learning, existing approaches struggle to invoke deep reasoning capabilities in multi-turn agents with long-horizon interactions. We propose DeepMiner, a novel framework that elicits such abilities by introducing high-difficulty training tasks and dynamic context window. DeepMiner presents a reverse construction method to generate complex but verifiable question-answer pairs from authentic web sources, which ensures the challenge and reliability of training data while injecting cognitive capabilities into multi-turn reasoning scenarios. We further design an elegant yet effective dynamic context management strategy for both training and inference, utilizing sliding window mechanisms while eliminating the dependency on external summarization models, thereby efficiently empowering the model to handle continuously expanding long-horizon contexts. Through reinforcement learning on Qwen3-32B, we develop DeepMiner-32B, which achieves substantial performance improvements across multiple search agent benchmarks. DeepMiner attains 33.5% accuracy on BrowseComp-en, surpassing the previous best open-source agent by almost 20 percentage points, and demonstrates consistent improvements on BrowseComp-zh, XBench-DeepSearch, and GAIA. Notably, our dynamic context management enables sustained interactions of nearly 100 turns *within standard 32k context length*, effectively addressing the context limitations that constrain existing multi-turn interaction systems.

## 1 Introduction

Recent advances in Reinforcement Learning with Verifiable Rewards (RLVR) have enabled significant breakthroughs in mathematical reasoning and code generation (DeepSeek-AI, 2025; Yang et al., 2025; Liu et al., 2025b). These methods empower large language models to exhibit sophisticated cognitive behaviors such as self-verification, backtracking, and subgoal decomposition in single-turn settings (Gandhi et al., 2025). The release of OpenAI's DeepResearch (OpenAI, 2025b) demonstrates that such cognitive capabilities can be successfully extended to multi-turn and long-horizon tasks, yet how to achieve this extension remains unknown. While existing open-source efforts have explored this direction through high-quality data generation (Li et al., 2025a; Tao et al., 2025; Gao et al., 2025; Wu et al., 2025) and specialized reinforcement learning algorithms (Song et al., 2025; Jin et al., 2025; Zheng et al., 2025), they have not achieved the same level of capability enhancement observed in single-turn settings. Substantial performance gaps persist compared to proprietary systems that maintain stable performance across dozens of interactions.

In this paper, we focus on search agents and identify two fundamental challenges that prevent existing approaches from effectively leveraging long-horizon interactions:

- **Insufficient Task Complexity.** Current datasets such as TriviaQA (Joshi et al., 2017), 2WikiMultihopQA (Ho et al., 2020), and HotpotQA (Yang et al., 2018), while involving multi-step reasoning, rely on structured Wikipedia data and permit success through shallow information retrieval. These tasks fail to elicit sophisticated cognitive behaviors such as verification, backtracking, and strategic planning that characterize expert-level reasoning.

- **Context Management Limitations.** Long-horizon interactions rapidly consume available context through accumulated tool responses, with a typical 32k context length supporting

only approximately 10-15 effective interaction turns. Current solutions primarily rely on summarization compression of tool outputs (Li et al., 2025a; Gao et al., 2025), but this approach introduces several limitations. First, summarization inevitably loses fine-grained information critical for precise reasoning. Second, additional summarization models increase system complexity and computational overhead. Third, summarization components cannot be integrated into end-to-end RLVR optimization, creating optimization blind spots.

To address these challenges, we present DeepMiner, a training framework for long-horizon multi-turn agents via constructing high-difficulty training tasks and managing continuously growing model contexts. For task complexity, we develop a reverse construction method that generates challenging QA pairs by synthesizing questions across multiple authentic web sources, with rigorous filtering to ensure genuine reasoning demands. Our primary contribution lies in addressing context management limitations through a dynamic sliding window strategy that selectively compresses distant tool responses while preserving assistant reasoning traces. Additionally, our strategy maintains access to original webpage content rather than compressed summaries, avoiding the information loss and optimization blind spots inherent in summarization-based approaches.

We implement DeepMiner on Qwen3-32B and evaluate the resulting model across multiple deep research benchmarks. DeepMiner-32B achieves 33.5% accuracy on BrowseComp-en, surpassing the previous best open-source agent nearly 20 percentage points. Consistent improvements are observed across BrowseComp-zh, XBench-DeepSearch, and GAIA. These advantages suggest that high-quality training data and effective context management can jointly facilitate the development of deep reasoning capabilities in long-horizon, multi-turn interactions. Notably, our dynamic context management enables sustained interactions 100 tool calls within standard 32k context length, demonstrating the effectiveness of our approach for extending the operational horizon of long-horizon reasoning agents.

## 2 COMPLEX QUESTION CONSTRUCTION

Training agents with deep reasoning capabilities in long-horizon interactions requires tasks that inherently demand extended exploration and sophisticated multi-step reasoning. Existing datasets predominantly enable success through simple retrieval or shallow reasoning patterns, failing to elicit the sophisticated cognitive strategies required in complex information-seeking tasks, such as verification, backtracking, and subgoal setting. We tackle this challenge through a reverse construction approach that produces complex QA pairs demanding extended reasoning across multiple authentic information sources.

Our approach grounds question generation in real-world entities and authentic web information, creating deliberately complex tasks that challenge expert-level reasoning abilities. Figure 1 illustrates our three-stage pipeline: entity-driven information collection, multi-source question generation, and strict quality filtering.

**Entity-driven Information Collection.**   We begin with entity selection from Wikipedia, targeting individuals with moderate visibility [1] to strike a balance between sufficient information availability and excessive familiarity. This range excludes both overly obscure figures lacking sufficient web presence and highly popular individuals whose information is already included in most models' parametric knowledge. For each selected entity, we conduct comprehensive information gathering through two primary search strategies: direct name queries for biographical information and news searches for recent developments, typically yielding dozens of potentially relevant web pages per entity. The collected sources undergo three-stage filtering to ensure source quality: entity correspondence verification, information complementarity assessment, and credibility validation. Entity correspondence verification compares each webpage against Wikipedia to eliminate entity confusion and confirm that pages refer to the right individual. Information complementarity assessment excludes sources that provide no substantial and unique knowledge.

**Question Generation.**   The question generation process takes multiple selected sources as input while deliberately excluding Wikipedia pages to force synthesis across distributed sources. We provide the LLMs with detailed demonstrations of complex reasoning patterns and explicitly constrain

---

[1] 1,000~10,000 page views over the last six months.

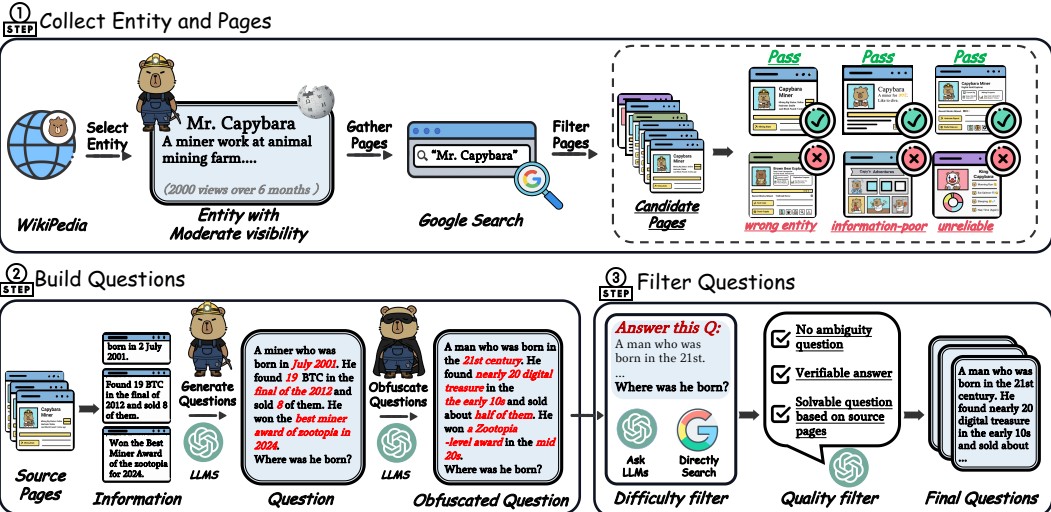

Figure 1: Overall pipeline for complex question construction.

each question to synthesize information from at least four distinct sources, requiring cross-document inference rather than simple fact extraction. To further increase complexity, generated questions undergo secondary obfuscation processing. This obfuscation process increases reasoning demands by replacing specific information with more generic descriptions, transforming concrete details into broader categorical terms while maintaining answer uniqueness. The obfuscated questions require the model to resolve generic descriptions through cross-document synthesis, while ensuring that questions remain solvable with sufficient exploration.

**Multi-stage Filtering.** Generated QA pairs undergo difficulty filtering and quality filtering to ensure both difficulty and reliability. *Difficulty filtering* ensures questions require extensive reasoning beyond current model capabilities through two mechanisms: direct search engine queries and zero-shot prompting of reasoning models. We then eliminate questions that find the entity or correct answer through either path, ensuring retained questions genuinely demand tool-assisted multi-step exploration. *Quality filtering* controls multiple aspects that could compromise reliability. We eliminate questions with the following attributes: 1) contains elements or expressions that may introduce interpretation ambiguity, 2) has evasive or ambiguous answers, and 3) the answer is not logically derivable from available evidence in the given reference documents. By eliminating ambiguity, filtering ambiguous answers, and verifying question validity, we ensure each QA pair provides a reliable training signal.

This reverse construction approach generates training tasks that demand extended multi-step reasoning and strategic planning across long-horizon scenarios. The requirement for multi-source information integration and deliberate obfuscation creates an environment where models must engage in genuine cross-document reasoning across extended interaction sequences, providing the challenging training substrate necessary for effective reinforcement learning optimization. Question examples and detailed construction prompts can be found in Appendix F.

## 3 REINFORCEMENT LEARNING WITH DYNAMIC CONTEXT WINDOW

### 3.1 DYNAMIC CONTEXT MANAGEMENT STRATEGY

**Empirical Analysis of Context Challenges.** Complex agent tasks typically require language models to engage in long-horizon interactions. However, in real-world scenarios, the extent of these interactions is fundamentally constrained by the model's maximum context length. To illustrate this challenge, we analyze error patterns of open-source models on the complex information-seeking task BrowseComp. As shown in Figure 2, even when setting the max context length to 128k, the majority of model failures occur when the maximum context length is reached. Case analysis reveals

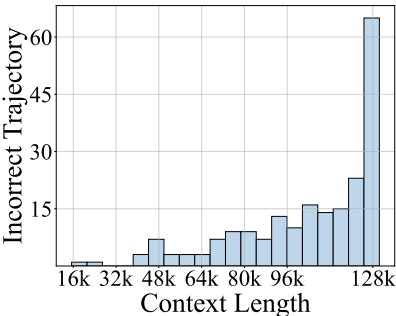 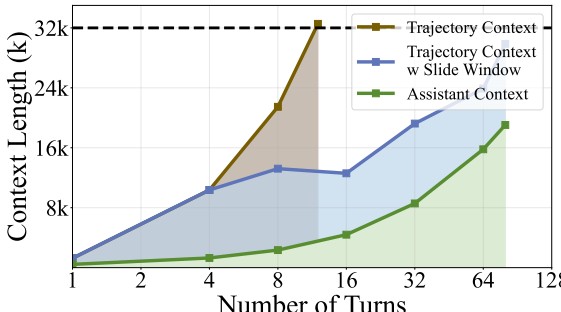

Figure 2: Preliminary experiments with gpt-oss-120b on the BrowseComp dataset. Left: Length distribution of incorrect trajectories. Right: Context length changes across rounds with/without sliding window. Without sliding window, tool responses grow exponentially and squeeze assistant context. With sliding window, tool length stays constant while assistant content grows normally.

that models often recognize they have not found the answer and attempt to continue searching, but are forced to terminate due to context constraints rather than task completion.

We further examine model output patterns under context limitations, with results presented in Figure 2. Under most used 32k length constraints, models typically reach context limits after only 10-15 turns, which proves insufficient for complex deep research tasks. Detailed analysis of context composition reveals that tool responses are typically 5-10 times longer than assistant responses, creating rapid context consumption that severely limits interaction depth. Further investigation into tool response shows that in most cases, current tool responses primarily influence the model's immediate next decision, with minimal impact on outputs at distant interaction positions. This observation suggests that while tool responses are essential for local decision-making, their long-term retention may not be necessary for maintaining coherent reasoning strategies across extended interactions.

**Sliding Window Mechanism.** Motivated by these findings, we design a context management strategy that selectively retains tool responses while preserving assistant reasoning traces. Let $\tau = \{q, a_1, t_1, a_2, t_2, \ldots, a_{T-1}, t_{T-1}, a_T\}$ represent a complete interaction trajectory, where $q$ denotes the user query, $a_i$ represents the $i$-th assistant response, and $t_i$ represents the corresponding tool response. We implement sliding window management with parameters window size $\mathcal{W}$ and slide size $\mathcal{S}$ to control tool response visibility.

As shown in Figure 3, the sliding operation triggers when the number of accumulated tools $|R_t|$ reaches $\mathcal{W}$. At this point, we identify the sliding boundary $b = \max(1, t - \mathcal{W} + \mathcal{S})$, then replacing tool responses $\{t_1, t_2, \ldots, t_{b-1}\}$ with placeholder token $\phi =$ "[Previous tool output skipped. Re-run tool if needed.]", while maintaining visibility of recent tool responses $\{t_b, t_{b+1}, \ldots, t_k\}$. This operation creates a modified trajectory where early tool responses are compressed while all assistant reasoning outputs remain intact, preserving the essential strategic reasoning and decision-making logic that enables coherent long-term planning. As shown in Figure 2, within the tool response sliding window mechanism, the search agent can even reach 100 turns within only 32k context.

**Training-Testing Consistency.** The sliding window mechanism introduces a new challenge: training models to operate effectively under dynamic context conditions. Direct training on complete trajectories would create a mismatch with inference behavior, where models encounter varying context states as sliding operations occur. To address this, we decompose each trajectory $\tau$ into multiple training sequences that reflect the different context states experienced during inference.

For a trajectory with $T$ tool calls, we generate $K = \lfloor \frac{T - \mathcal{W}}{\mathcal{S}} \rfloor + 1$ training sequences. The first sequence $\tau^{(1)}$ contains complete initial context with all assistant responses trained. For subsequent sequences $k > 1$, we construct training instances where early tool responses are replaced with placeholders according to the sliding boundary, while recent context remains within the active window.

To prevent optimization conflicts, we ensure each assistant response is trained exactly once across all sequences through careful masking:

$$M_i^{(k)} = \begin{cases} 0 & \text{if } i < \mathcal{W} + (k-2) \cdot \mathcal{S} + 2 \\ 1 & \text{otherwise} \end{cases}$$

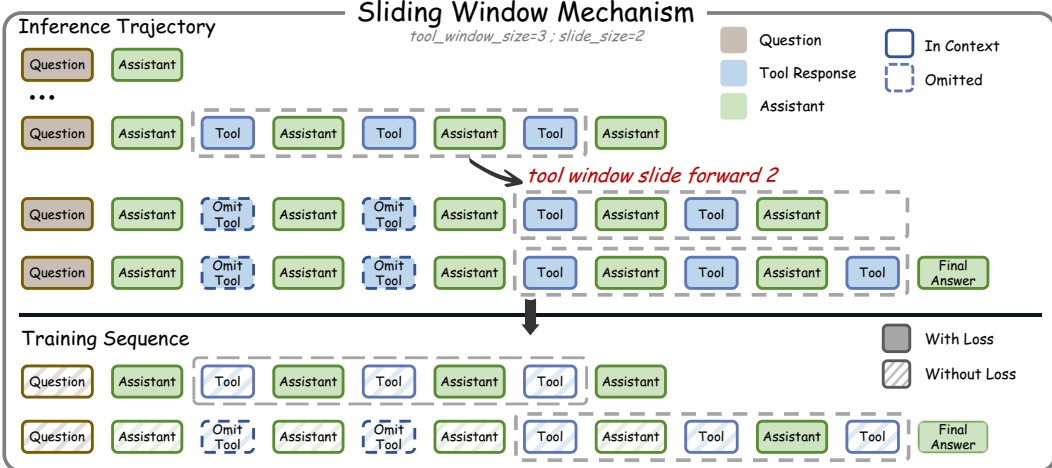

Figure 3: Sliding window mechanism for dynamic context management. Upper: inference trajectory with sliding window (size=3, slide=2) where older tool responses are replaced by omission tokens while assistant outputs are preserved. Lower: corresponding training sequences where only newly generated responses receive gradient updates while previous outputs serve as fixed context.

This training strategy ensures models learn to reason effectively under the same dynamic context conditions they encounter during inference, maintaining consistency between training and deployment scenarios while enabling scalable optimization on arbitrarily long interaction sequences.

### 3.2 COLD START

To establish foundational tool-calling and long-horizon reasoning capabilities, we employ supervised fine-tuning as a cold-start phase. We design a pipeline to generate action trajectories through powerful language models, where the sliding window mechanism is also applied during trajectory generation to eliminate turn limitations imposed by context constraints. We filter trajectories with incorrect answers or excessive length, retaining high-quality examples that demonstrate effective tool usage and multi-step reasoning. During training, we employ the multi-sequence construction method described in Section 3.1 to ensure training-testing consistency. This cold-start phase provides models with essential capabilities for structured tool interaction and coherent reasoning under dynamic context conditions, establishing the foundation for subsequent reinforcement learning optimization.

### 3.3 REINFORCEMENT LEARNING TRAINING

**Trajectory-level Advantage with Sequence-level Training.** We employ Group Relative Policy Optimization (Shao et al., 2024) as our reinforcement learning algorithm, and adapt the advantage computation to work with our sliding window context management. The key challenge lies in reconciling trajectory-level reward signals with sequence-level training requirements imposed by sliding window processing.

For each question $q$, we generate $G$ complete trajectory rollouts using the sliding window mechanism during inference. Rewards are computed at the trajectory level based on the final answer, and advantages are calculated through group-relative comparison among these $G$ trajectories:

$$\hat{A}_i = \frac{R_i - \text{mean}(\{R_j\}_{j=1}^G)}{\text{std}(\{R_j\}_{j=1}^G)}$$

Our modification lies in advantage propagation: each trajectory $\tau_i$ is decomposed into multiple training sequences $\{\tau_i^{(1)}, \tau_i^{(2)}, \ldots, \tau_i^{(K_i)}\}$ following the sliding window protocol. The trajectory-level advantage $\hat{A}_i$ is then propagated to all training sequences derived from that trajectory, ensuring each sequence receives the same advantage signal regardless of its specific context state. This approach maintains the group-relative advantage computation that drives effective policy learning while enabling training under the dynamic context conditions that models encounter during inference.

**Reward Design.** We employ a straightforward reward design where an LLM judge evaluates whether the final answer matches the ground truth, assigning a binary reward of 1 for correct answers and 0 for incorrect ones. This binary signal provides clear optimization targets while avoiding complex reward engineering that might introduce training biases or instabilities in the long-horizon setting.

# 4 EXPERIMENTS

## 4.1 EXPERIMENTAL SETUPS

**Tool Configuration.** Our agent operates with three tools: *web_search* returns top-10 web search results with titles, URLs, and snippets; *fetch* retrieves webpage content in Markdown format with pagination, enabling efficient navigation of long documents; and *find* performs in-page search to locate specific information within webpages. This tool suite supports comprehensive information retrieval while maintaining computational efficiency.

**Training Details.** We adopt Qwen3-32B Yang et al. (2025) as our base model, with the thinking mode enabled. We employ a two-stage training strategy. For SFT, we train on approximately 3,000 high-quality trajectories using a batch size of 256 and learning rate of 1e-5. The RL phase uses around 4,000 questions with a batch size of 32 and learning rate of 2e-6. During on-policy RL training, we generate 8 rollouts per question for policy optimization, with a maximum trajectory length of 40k tokens and turn limit of 60. To manage context length, we set the tool window size to 5 with slide size of 3. All training is implemented using the VERL framework.

**Evaluation.** We evaluate DeepMiner on four challenging deep research benchmarks: BrowseComp (Wei et al., 2025), BrowseComp-zh (Zhou et al., 2025), XBench-DeepSearch (Chen et al., 2025), and GAIA (Mialon et al., 2023). We use complete test sets for main experiments and a BrowseComp subset for detailed analysis. Our evaluation employs the following configuration: temperature of 0.6, top-p of 0.9, maximum of 100 interaction turns. For context management, we set the sliding window size to 5 with a slide size of 3. All results are evaluated using ChatGPT-4o-Latest as the judge model, with evaluation prompts provided in Appendix E.

**Baselines.** We compare DeepMiner against three categories of systems for comprehensive performance evaluation. First are commercial systems, primarily including leading products such as OpenAI DeepResearch (OpenAI, 2025b), Metaso DeepResearch (Metaso, 2025), and Kimi Researcher (Team et al., 2025). Second are advanced general models, including OpenAI-o3 (OpenAI, 2025a), Claude-4-Sonnet (anthropic, 2025), DeepSeek-R1 (DeepSeek-AI, 2025), DeepSeek-V3.1 (DeepSeek Team, 2025), Kimi-K2 (Team et al., 2025), GLM-4.5 (Zeng et al., 2025). Finally, recent open-source work includes Webshaper (Tao et al., 2025), ASearcher (Gao et al., 2025), WebDancer (Wu et al., 2025), WebSailor (Li et al., 2025a), WebExplorer (Liu et al., 2025a), and DeepDive (Lu et al., 2025b). We adopt the official results reported in their papers.

## 4.2 MAIN RESULTS.

We evaluate DeepMiner across four challenging deep research benchmarks. As shown in Table 1, DeepMiner achieves substantial improvements over existing open-source agents while approaching the performance levels of leading proprietary systems.

DeepMiner achieves substantial improvements over existing open-source agents, with particularly impressive results on the challenging BrowseComp benchmarks. On BrowseComp-en, DeepMiner-32B reaches 33.5% accuracy, substantially outperforming all the previous open-source agents and notably exceeding DeepSeek-V3.1-671B, a model over 20 times larger. Notably, even our SFT model demonstrates strong performance, significantly outperforming most existing open-source agents and highlighting the effectiveness of our reverse construction method for generating high-quality training data. The performance gains extend consistently across BrowseComp-zh, XBench-DeepSearch, and GAIA. This consistent performance across benchmarks validates both the generalizability of our approach and the diversity of our automatically constructed training data.

The progression from supervised fine-tuning to reinforcement learning demonstrates substantial and consistent improvements across all evaluated benchmarks, confirming the effectiveness of our train-

| Model | BrowseComp | BrowseComp-zh | Xbench-DS | GAIA |
|---|---|---|---|---|
| *Commercial Agents* | | | | |
| OpenAI DeepResearch | 51.5 | 42.9 | - | 67.4 |
| Metaso DeepResearch | 12.0 | 45.4 | 64.0 | - |
| Kimi Researcher | - | - | - | 69.0 |
| *General LLMs with tools* | | | | |
| OpenAI-o3 | 49.7 | 58.1 | 66.7 | 70.5 |
| Claude-4-Sonnet | 12.2 | 29.1 | 64.6 | 68.3 |
| DeepSeek-R1 | 8.9 | 35.7 | 55.0 | - |
| Kimi-K2-Instruct-1T | 14.1 | 28.8 | 50.0 | 57.7 |
| GLM-4.5-355B | 26.4 | 37.5 | 70.0 | 66.0 |
| DeepSeek-V3.1-671B | 30.0 | 49.2 | 71.2 | 63.1 |
| *Open-source Agents* | | | | |
| WebShaper-72B | - | - | - | 60.0 |
| SearchR1-32B | - | - | 28.6 | 19.5 |
| ASearcher-32B | - | - | 42.1 | 52.8 |
| WebDancer-32B | 3.8 | 18.0 | 39.0 | 51.5 |
| WebSailor-72B | 12.0 | 30.1 | 55.0 | 55.4 |
| DeepDive-32B | 14.8 | 25.6 | 50.5 | - |
| WebExplorer-8B | 15.6 | 32.0 | 53.7 | 50.0 |
| *Our Agents* | | | | |
| DeepMiner-32B-SFT | 21.2 | 28.0 | 53.0 | 54.4 |
| DeepMiner-32B-RL | 33.5 | 40.1 | 62.0 | 58.7 |

Table 1: Performance comparison on deep research benchmarks.

ing methodology. RL optimization provides significant gains over SFT model: +12.3 percentage points on BrowseComp-en, +12.1 points on BrowseComp-zh, +9.0 points on XBench-DeepSearch, and +4.3 points on GAIA. The magnitude of improvement is especially pronounced on the most challenging benchmarks—BrowseComp-en and BrowseComp-zh—where tasks demand sophisticated reasoning strategies. This pattern suggests that reinforcement learning with our generated dataset effectively enhances the model's ability to develop multi-step reasoning strategies and maintain coherent search policies across extended interactions.

### 4.3 CONTEXT MANAGEMENT EFFICIENCY ANALYSIS.

To validate the efficiency of our sliding window mechanism, we conduct experiments on both training-free settings and training settings.

| Strategy | Detail Availablity | End to End Optimization | 32k | 64k | 128k |
|---|---|---|---|---|---|
| Vanilla | ✓ | ✓ | 9.0 | 23.7 | 30.3 |
| Summary | ✗ | ✗ | 10.0 | 25.3 | 31.6 |
| Sliding Window | ✓ | ✓ | 33.3 | 33.3 | 33.3 |

Table 2: Comparison of different context management strategies across multiple dimensions. All evaluations are conducted on GPT-OSS-120B. DeepMiner's performance remains stable, as it nearly reaches the maximum of 100 rounds within 32k context length.

**Training-free Setting** We conduct training-free evaluation using GPT-OSS-120B across three context management strategies: vanilla approach (no context management), external summarization, and our sliding window method. Table 2 shows that our sliding window method achieves superior context utilization efficiency while maintaining critical technical advantages. Our approach demonstrates remarkable context efficiency: DeepMiner achieves 33.3% accuracy on BrowseComp using only 32k context, exceeding the performance of alternative methods that require 128k context length. These results confirm that dynamic sliding window management enables more efficient context utilization while providing fundamental technical advantages. The superior performance

achieved in this training-free evaluation demonstrates the inherent efficiency of our approach for enhancing long-horizon reasoning capabilities.

| Strategy | BrowseComp | BrowseComp-zh | Xbench-DS | GAIA |
|---|---|---|---|---|
| Truncation | 19.7 | 62.8 | 25.6 | 41.6 | 47.0 | 26.9 | 44.7 | 15.3 |
| Sliding Window | 21.2 | 36.6 | 28.0 | 26.4 | 53.0 | 21.1 | 54.5 | 13.6 |

Table 3: Performance comparison between Vanilla Context with Truncation and Sliding Window under the same training setup. Each cell reports "Accuracy | Average number of reasoning turns".

**Training Setting**  We further validate the sliding-window strategy in the training setting by comparing it against a truncation-based context management baseline using the DeepMiner dataset. For the truncation baseline, whenever the full dialogue is about to exceed the model's maximum context length, we remove the oldest turns to allow the next reasoning step to proceed. This experiment is conducted under a fully fair setup, using the same DeepMiner dataset, the same base model, and the same inference strategy. The results are shown in Table 3, which indicates that the sliding window model consistently outperforms the truncation model across all benchmarks, demonstrating that the dynamic context window yields clear and measurable gains beyond what the DeepMiner dataset alone can provide. Moreover, the sliding-window model requires substantially fewer reasoning turns on every benchmark. This shows that the benefits of the sliding-window strategy do not come from simply enabling longer interactions; instead, it constitutes a fundamentally more effective context management mechanism.

## 4.4 DETAILED ANALYSIS

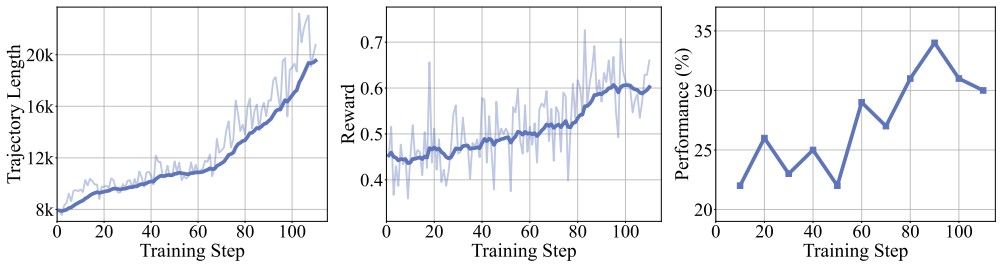

Figure 4: Training Dynamics of DeepMiner Reinforcement Learning.

**Training Dynamics.**  Figure 4 illustrates the training dynamics of our reinforcement learning process. As shown in the figure, our trajectory length steadily increases with training steps while remaining within the 40k trajectory length limit. As analyzed in Section 3.1, this indicates continuous growth of assistant context, reflecting the model's progressively enhanced capabilities in multi-step reasoning and complex search strategies. This improvement is reflected in the training rewards, which exhibit a sustained growth trend throughout the process, gradually increasing from 0.45 to 0.60 and converging after 90 steps, indicating that our tasks are of sufficient difficulty for effective policy learning. We select the checkpoint at step 90 as the model for the main experiments. The in-domain rewards translate to advantages on evaluation benchmarks, where BrowseComp performance improves from 22% to 33.5%, confirming that our reverse-constructed tasks can stimulate the model to perform deep cognitive behaviors in general long-horizon multi-turn scenarios.

**Context Scaling.**  We analyze how DeepMiner scales with the tool-call budget and the maximum context length on BrowseComp. As shown in the left subplot of Figure 5, DeepMiner's performance increases steadily with larger tool-call budgets, surpassing DeepSeek-V3.1-671B at around 60 calls. At 100 calls, performance reaches 33.5, clearly exceeding the majority of open-source agents and approaching the level of leading proprietary systems, which indicates that tool-call capacity is a central determinant of agent capability. The right subplot reports DeepMiner's performance under

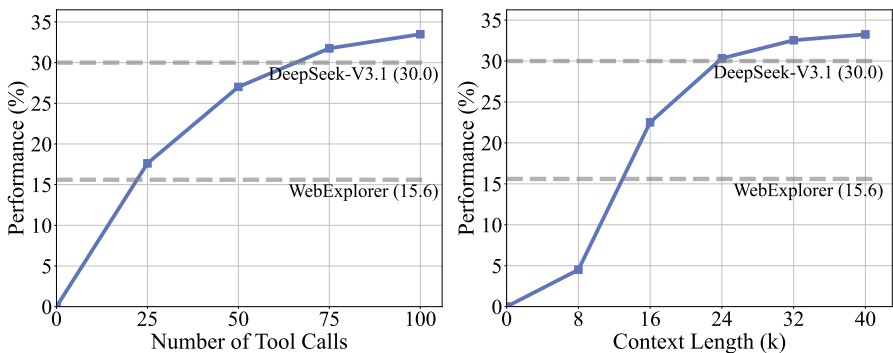

Figure 5: Scaling on tool call budget and context length.

varying context lengths. With a commonly used 32k context length, performance approaches 33.0 – nearly double that of other open-source agents under the same settings. This pattern highlights not only the efficiency of DeepMiner's context management strategy but also its ability to sustain close to 100 rounds of tool interactions within a constrained context space, demonstrating that careful management of context may be more effective than simply expanding capacity.

**Data Efficiency.** To validate the efficiency and effectiveness of our constructed QA pairs, we compare the DeepMiner dataset with HotpotQA (Yang et al., 2018), a widely used dataset in search agent training (Jin et al., 2025). We conduct the same cold-start process as in our main experiments to construct an SFT model based on HotpotQA training data. As shown in Table 4, the model trained on HotpotQA demonstrates inferior performance compared to the DeepMiner SFT model. The HotpotQA-trained model achieves only 15.6% on BrowseC-

| Data | BC | BCZH | Xbench | GAIA |
|---|---|---|---|---|
| HotpotQA | 15.6 | 21.8 | 47.0 | 51.4 |
| DeepMiner | 21.2 | 28.0 | 53.0 | 54.4 |

Table 4: Comparison between HotpotQA-SFT and DeepMiner-SFT.

omp, while the DeepMiner-trained model reaches 21.2%. This performance gap demonstrates that conventional multi-hop datasets are insufficient to elicit the cognitive behaviors required for complex web agent tasks, validating the necessity of our challenging data construction approach. Additionally, we compare the data effectiveness between concurrent work WebShaper (Tao et al., 2025) and DeepMiner. As shown in Appendix C, DeepMiner achieves superior performance, further confirming its data effectiveness.

**Hyperparameter Analysis.** We conducted an additional experiment to evaluate how window size $\mathcal{W}$ and slide size $\mathcal{S}$ affect model performance. While DeepMiner was optimized under a fixed parameter ($\mathcal{W} = 5, \mathcal{S} = 3$), we assess its zero-shot generalization across different inference-time settings on a 100-sample BrowseComp subset. Table 5 reports the results of DeepMiner under $\mathcal{W}$ from 3 to 6 and $\mathcal{S}$ from 1 to 5. The model exhibits significant robustness, maintaining performance predominantly in the $30\% - 38\%$ range across diverse settings. This phenomenon suggests that DeepMiner does not overfit

| $\mathcal{W} \setminus \mathcal{S}$ | 1 | 2 | 3 | 4 | 5 |
|---|---|---|---|---|---|
| 3 | 32 | 37 | - | - | - |
| 4 | 33 | 30 | 38 | - | - |
| 5 | 33 | 33 | 34 | 31 | - |
| 6 | 31 | 32 | 34 | 34 | 35 |

Table 5: The results of DeepMiner under different window sizes and slide sizes. "-" indicates the result is unavailable.

to the specific context window parameters but acquires a generalized and adaptive capability to reason over varying context management policies. Such robustness enables flexible deployment, allowing $\mathcal{W}$ and $\mathcal{S}$ to be adjusted to balance the cost and context retention without retraining.

**Model Size Scaling.** To assess how model capacity affects deep search performance, we evaluate Deep-Miner across the Qwen3 family (4B, 8B, 14B, and 32B). As shown in Table 6, performance scales consistently with model size on all benchmarks. Interestingly, even the smaller models achieve reasonably strong results on the challenging BrowseComp task, highlighting the efficiency of the DeepMiner data in enabling effective reasoning behaviors at relatively small scales. These findings indicate that DeepMiner provides a robust training signal across a wide range of model capacities.

| Size | BC | BCZH | Xbench | GAIA |
|------|------|------|--------|------|
| 4B | 14.3 | 13.8 | 35.0 | 38.8 |
| 8B | 15.7 | 18.3 | 35.0 | 43.7 |
| 14B | 20.0 | 27.0 | 47.0 | 53.4 |
| 32B | 21.2 | 28.0 | 53.0 | 54.4 |

Table 6: Scaling on model size with Qwen3 family.

.

## 5 RELATED WORK

**Reinforcement Learning with Verifiable Rewards.** Currently, Reinforcement Learning with Verifiable Rewards (RLVR) has become the standard method for training LLMs (DeepSeek-AI, 2025; Team et al., 2025; Yang et al., 2025). Various RL algorithms, such as GRPO (Shao et al., 2024), have been proven to demonstrate superior performance across multiple complex tasks. Recently, several works have explored improving model capability using synthetic data combined with RLVR (Guo et al., 2025), as well as applications to web agents (Lu et al., 2025b; Wu et al., 2025; Tao et al., 2025; Jin et al., 2025; Wang et al., 2025). However, the unique application scenarios of web agents introduce new challenges, such as sparse rewards and context limitations, making it difficult for RLVR methods to achieve their expected benefits. To address this, rather than focusing on constructing new training mechanisms or rewards for reinforcement learning (Yu et al., 2025; Liu et al., 2025b; Lu et al., 2025a), we leverage a dynamic context window mechanism to substitute the context management framework of the ReAct Yao et al. (2023) used in most multi-turn RL. This extends the application scenarios of RLVR without significantly increasing computational costs, achieving efficient long-horizon RL training.

**Deep Research Agents.** Deep research agents are LLM-based systems that autonomously leverage search engines, web browsing, and various tools to accomplish complex research tasks through multi-step deep reasoning processes (Xi et al., 2025). Proprietary systems like OpenAI's DeepResearch (OpenAI, 2025b) and others (Google, 2025; Claude Team, 2025) have exhibited unprecedented ability in complex tasks, matching or exceeding human experts. However, their closed architectures and inaccessible training data hinder reproducibility and collaborative research. Concurrently, the open-source community has achieved significant progress in fundamental tasks (Zhang et al., 2025; Li et al., 2025b; OpenPangu Team, 2025). Recent works have shifted toward deep research tasks that demand complex nonlinear reasoning capabilities. Researchers have developed various data synthesis approaches for agent optimization (Liu et al., 2025a; Lu et al., 2025b; Tao et al., 2025). Some other works design context management strategies, such as summarization (Li et al., 2025a; Gao et al., 2025), for deeper interactions. Despite these advantages, a huge gap persists - even state-of-the-art open-source models achieve less than 20% accuracy on the BrowseComp, far below proprietary systems. Meanwhile, DeepMiner significantly narrows this gap through reverse-constructed complex training tasks and dynamic context window.

## 6 CONCLUSION

In this work, we present DeepMiner to build deep research agents with cognitive behaviors that can effectively handle sustained long-horizon interactions. DeepMiner addresses two fundamental limitations for web agent training: insufficient training task complexity through reverse construction of verifiable QA pairs from authentic web sources, and context explosion through dynamic context management without external summarization models. DeepMiner preserves complete assistant reasoning traces while compressing distant tool responses, enabling interactions that exceed 100 turns within 40k contexts. DeepMiner-32B achieves 33.5% on BrowseComp, nearly doubling the performance of prior state-of-the-art open-source agents. DeepMiner represents a fundamental shift from context-limited, shallow reasoning to unbounded, deep exploration.

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

## A  THE USE OF LARGE LANGUAGE MODELS

We utilized LLMs to aid and polish writing.

## B  ETHICAL CONSIDERATIONS

Our training data collection process may inadvertently include personal information from publicly available sources. To mitigate privacy risks, we collect data exclusively from public webpages like Wikipedia, filtering out irregular websites and social media platforms to avoid overly private content. We will release our dataset under strict access controls, requiring formal approval for academic use only. Before public release, we will perform comprehensive anonymization, replacing names and other identifying information to ensure no real personal privacy is compromised. We acknowledge that our models could potentially be misused. To address this concern, we will implement a rigorous review process for all requests to access our open-source model weights, ensuring they are used solely for legitimate research and educational purposes.

## C  COMPARATIVE DATA EFFECTIVENESS WITH CONCURRENT WORK

| Question Source | BrowseComp | BrowseComp-zh | Xbench-DS | GAIA |
|---|---|---|---|---|
| WebShaper | 13.7 | 22.1 | 50.0 | 48.5 |
| DeepMiner | 16.3 | 23.9 | 50.0 | 55.3 |

Table 7: Performance comparison on models trained with questions from WebShaper and Deep-Miner. We report the performance of SFT models trained on data delivered from 500 questions.

To further validate DeepMiner's data effectiveness, we conduct a controlled comparison with Web-Shaper (Tao et al., 2025), which provides 500 open-source questions. We evaluate data quality from two perspectives: (1) difficulty comparison: using a strong open-source model GPT-OSS-120B, we directly evaluate questions from both datasets. GPT-OSS-120B achieves 37% accuracy on DeepMiner but 61% on WebShaper. It demonstrates that DeepMiner samples are consistently more challenging, implying more room for RL improvement. (2) SFT performance as a proxy for data quality: we compare downstream training effectiveness under controlled conditions: identical base model (Qwen3-32B), training pipeline, SFT data collection, and question size (500). As shown in Table 7, the DeepMiner and WebShaper model achieves 16.3% vs. 13.7% on BrowseComp and 55.3% vs. 48.5% on GAIA. These results show that DeepMiner yields stronger generalization and better data efficiency.

## D  ENHANCED TOOL SUITE

Our agent operates with three core tools designed for fine-grained web exploration:

1. **Web Search** takes text queries as input and returns document titles, URLs, and snippets from search results.

2. **Fetch** retrieves webpage content given a URL. Unlike existing approaches that truncate content or apply external summarization, our fetch tool implements paginated browsing that mimics human web navigation, allowing the model to assess initial content and decide whether to continue reading or exit.

3. **Find** enables in-page keyword search for lengthy webpages. The model can locate relevant information sections and the surrounding context before deciding which portions warrant detailed examination.

This design addresses a fundamental limitation where existing web agents suffer information loss through hard truncation or external summarization. By preserving complete information access while providing navigation flexibility, our tool suite enables fine-grained control over information gathering across extended interaction sequences.

# E  JUDGMENT TEMPLATES

Below is the judgment template we used.

---

**Judgment Template**

Judge whether the following [response] to [question] is correct or not based on the precise and unambiguous [correct_answer] below.

[question]: question

[response]: response

Your judgment must be in the format and criteria specified below: extracted_final_answer: The final exact answer extracted from the [response]. Put the extracted answer as 'None' if there is no exact, final answer to extract from the response.

[correct_answer]: answer

reasoning: Explain why the extracted_final_answer is correct or incorrect based on [correct_answer], focusing only on if there are meaningful differences between [correct_answer] and the extracted_final_answer. Do not comment on any background to the problem, do not attempt to solve the problem, do not argue for any answer different than [correct_answer], focus only on whether the answers match.

correct: Answer 'yes' if extracted_final_answer matches the [correct_answer] given above, or is within a small margin of error for numerical problems. Answer 'no' otherwise, i.e. if there if there is any inconsistency, ambiguity, non-equivalency, or if the extracted answer is incorrect.

---

# F  DETAILS OF QUESTION CONSTRUCTION PIPELINE

## F.1  MODELS USED TO SYNTHESIZE DATA

We primarily use Qwen3-235B-A22B-Thinking-2507 and DeepSeek-R1-0528 to construct questions. In the document filtering stage, for cost and performance considerations, we only use Qwen3-235B-A22B-Thinking-2507 to filter web documents. When generating questions, we utilize Qwen3-235B-A22B-Thinking-2507 and DeepSeek-R1-0528 to enhance question diversity. For question filtering, we prompt both models with each question separately, and if either mentions the answer, we drop that question.

## F.2  DOCUMENT FILTERING

After collecting documents, each of them undergoes a three-stage filtering to ensure quality, including entity correspondence verification, information complementarity assessment, and credibility validation. Detailed templates are as follows:

---

**Document Filter Template (Entity Correspondence Verification)**

[Task]
Evaluate the quality and suitability of the provided webpage for further processing. Determine whether it is directly related to the specific entity described by the provided Wikidata.

---

[Wikidata]
{wikidata}

[Web Content]
**Link**: {link}
**Content**: {content}

[Evaluation Criteria]
Assess the entity relevance of this page:
- Verify that the webpage explicitly and accurately focuses on the exact entity described by Wikidata.
- Confirm it's not about another individual with the same or similar name.
- The main content should be text and accessible by request, avoiding web pages that are mainly about videos, images, pdf, files, etc.

[Output Format]
Provide a detailed analysis and output your final assessment as follows:
**Acceptable**: {{yes or no}}

---

### Document Filter Template (Information Complementarity Assessment)

[Task]
Evaluate the quality and suitability of the provided webpage for further processing. Determine whether it contains specific, valuable, and credible information about the entity.

[Wikidata]
{wikidata}

[Web Content]
**Link**: {link}
**Content**: {content}

[Evaluation Criteria]
Assess the information complementarity of this page: - Contains substantial new information not present in Wikidata.
- Confirm the webpage provides specific details, insights, or data that enrich or supplement the information present in the Wikidata entry.
- Check for unique or exclusive information, direct quotes, primary source material, or additional context not covered by Wikidata.
- The main content should be text and accessible by request, avoiding web pages that are mainly about videos, images, pdf, files, etc.

[Output Format]
Provide a detailed analysis and output your final assessment as follows:
**Acceptable**: {{yes or no}}

---

### Document Filter Template (Credibility Validation)

[Task]
Evaluate the quality and suitability of the provided webpage for further processing. Determine whether it is formal, authoritative, believable, and trustworthy.

[Wikidata]
{wikidata}

[Web Content]
**Link**: {link}
**Content**: {content}

[Evaluation Criteria]
Assess the credibility of this page: - Evaluate the formality, reliability, and trustworthiness of the content.
- Consider the source's authority (e.g., official organizations, reputable media outlets, recognized academic institutions, professional profiles).
- Avoid informal sources such as social media platforms, forums, or websites with questionable authenticity.
- The main content should be text and accessible by request, avoiding web pages that are mainly about videos, images, pdf, files, etc.

[Output Format]
Provide a detailed analysis and output your final assessment as follows:
**Acceptable**: {{yes or no}}

## F.3 QUESTION GENERATION

We present the core prompt for question generation. Some question examples and context cases have been omitted.

---

**Question Generation Template**

You will be provided with multiple documents from different sources about an entity. You need to generate extremely high-difficulty retrieval questions to test users' search abilities using search engines only. The purpose is to test users' search engine retrieval skills. Users will only see the question text and must find answers through web searches - they have no access to the provided documents.

Core Requirements:

1. Cross-source integration: Questions must combine information from at least 4 different source documents, and this question cannot be answered solely through the content of any single document.

2. Difficulty through hard-to-retrieve features: The questions should use characteristics that are harder to search for directly, avoiding direct keyword approaches.
- Each clue should be necessary and sufficient, avoid redundant descriptive details
- Heavily reduce use of easily searchable proper nouns, job titles, and location names
- Use numerical ranges, distances, time periods, and other measurable constraints rather than descriptive narratives

3. Feature obfuscation: Obfuscate features appearing in questions to increase difficulty:
- "1905" $\rightarrow$ "early 20th century"
- "born in Manhattan" $\rightarrow$ "born in New York"

4. Answer specificity: The answer cannot be the entity itself, but should be related attributes, dates, people, etc. The answer must maintain uniqueness and specificity - avoid questions like "Who are this person's colleagues?" because there may be multiple valid answers.

5. Unique, verifiable answers: Each question must have one clear, precise answer that can be definitively verified. Ensure the combination of features is unique enough that only one entity can satisfy all the specified characteristics, minimizing the possibility of multiple

---

valid answers.

Example Questions:
(omitted example questions)

Ket Entity: {entity name}

Documents: {documents}

Now, you need to generate at least 2 questions.

## F.4  QUESTION FILTERING

We present the core prompt used by the most effective filter during question filtering, which identifies solvable questions based on source pages and ensures the answer is derivable from the given reference documents. Some question examples and context cases have been omitted.

---

**Question Filter Template**

You need to determine whether the given question-answer pair can be verified based on the provided document content. This is a retrieval question verification task, focused on ensuring the question does not mislead searchers to find incorrect answers.

[Question]
{question}

[Provided Answer]
{answer}

[Document Content]
{document}

## Task Background

These are complex retrieval questions with the following characteristics:
- Use multiple constraints combined to lock onto a unique answer
- Deliberately obscure certain information to increase retrieval difficulty
- Searchers need to find relevant documents through search engines, then reason to get the answer
- Your task is to ensure the question does not mislead searchers to find wrong answers

## Core Verification Principles

### 1. Multi-Condition Combination Principle
Individual constraints can be vague, but all constraints combined must point to a unique answer. Don't reject questions just because a single condition isn't precise—evaluate the overall combined effect.

### 2. Factual Accuracy Principle
Every description in the question must be objectively factually correct. Allow reasonable transitions from specific to abstract, but absolutely prohibit any factual errors.

### 3. Temporal Context Principle
Documents often contain time-sensitive statements from their writing period. Questions must account for the document's temporal context, not assume current validity.

---

## Verification Process

### Step 1: Temporal Context Analysis
- Identify when the document was written
- Note any time-sensitive claims ("currently", "now", "recently", "latest", "this year")
- Check if the question inappropriately transfers temporal language

### Step 2: Individual Constraint Verification
- Verify each constraint type following the guidelines above
- Look for factual errors, unreasonable abstractions, or category confusion

### Step 3: Answer Verification
- Confirm the document contains information needed to answer the question
- Verify the provided answer is consistent with document content

### Step 4: Risk Assessment
- Assess if the question could mislead searchers to find wrong answers
- Identify potential ambiguities that could cause confusion

## Final Verification Checklist

- [ ] Are there any factual errors in the question?
- [ ] Does the document contain the core information needed to answer the question?
- [ ] Are there any temporal context mismatches?
- [ ] Do time-sensitive claims in the question match the document's temporal context?

## Output Format

Provide your analysis covering:
1. **Temporal Context Analysis**: Document timeframe and time-sensitive claims
2. **Individual Constraint Verification**: Check each constraint for accuracy
3. **Answer Verification**: Confirm the document supports the answer
4. **Risk Assessment**: Identify potential issues (factual errors, ambiguity, temporal mismatches)

**Final Judgment**: Acceptable / Unacceptable

**Reasoning**: [2-3 sentences explaining your decision, highlighting key factors and any temporal context issues]

## F.5 GENERATED QUESTIONS

We show several representative questions below. We present the question, the core entity, and the answer.

---

Question Examples

### Question Example 1

**Question:**
A journalist who graduated from an Ivy League institution in the 1990s and later worked as a White House correspondent covering national security topics joined a major American newspaper between 2018 and 2020. This journalist previously worked at a political news organization where they served as an editor focusing on both domestic executive branch coverage and international affairs. During their tenure at this political publication, they wrote a piece with a provocative headline suggesting a connection between a major world leader and a specific presidential candidate. In a different role at a weekly news

---

magazine between 2009 and 2015, this same journalist served in a senior position covering international relations while also holding a deputy leadership role in that publication's Washington office. What is the birth month and day of this journalist?
**Entity:** Michael Crowley (journalist)
**Answer:** April 1

**Question Example 2**

**Question:**
An academic who transitioned from directing a public affairs institute in the Midwest to a Pennsylvania university position co-authored a book about religious themes in the late 2000s with a Pulitzer Prize-winning author. This academic appeared on a PBS program in early 2014 and published a memoir in the early 2020s that was reviewed by someone surnamed Begiebing. The memoir explores his grandfather's gambling problems. What New England city did this academic leave for college?
**Entity:** J. Michael Lennon
**Answer:** Fall River

**Question Example 3**

**Question:**
A basketball player who missed an entire season in the late 2010s due to a leg injury later earned a major academic honor at the Division I level between 2020 and 2023. This player's father competed at a university whose basketball team reached the NCAA tournament in the early 1990s. The player transferred to an Atlantic Coast Conference school for their final collegiate season. What month and day was this player born?
**Entity:** Ben Vander Plas
**Answer:** September 19

## G   CASE STUDY

The following is a case trajectory of DeepMiner on BrowseComp.

**A Case Trajectory on BrowseComp**

**Question**

I am looking for the name of a historical place that meets the following criteria: 1. As of 2023, the place is located in the capital city of a country. 2. It is situated beside a river as of 2023. 3. Its construction began between 1830 and 1860 (inclusive). 4. The construction was completed between 1870 and 1880 (inclusive). 5. The thickness of its walls ranges from 0.5 to 0.9 meters (inclusive). 6. It was acquired by the government of the country between 1980 and 1990(inclusive). 7. This place was once damaged by a tornado between 1880 and 1890(inclusive). 8. It also suffered damage from an earthquake between 1890 and 1900(inclusive). 9. The president of the country at the time of its acquisition was born between 1920 and 1935(inclusive).

**Step 1**

**Analysis:**
Search results not obvious. Perhaps we need to think of specific known historical places that were damaged by tornado and earthquake.
Tornado damage to a historical site in late 1800s: maybe the Alamo? No tornado.
Earthquake early 1900s: maybe the 1906 San Francisco earthquake. Tornado damage earlier? Not likely.
Capital city beside a river: many.
Acquired by government between 1980-1990: maybe nationalization of a historic house.
President at acquisition time born 1920-1935: which president? For example, US President Ronald Reagan (born 1911) no. US President George H.W. Bush (born 1924). If acquisition happened during his presidency (1989-1993). If acquisition was 1989, president George

H.W. Bush (born 1924). So US capital Washington D.C. Beside Potomac River. Historical place built 1830-1860 start, completed 1870-1880, walls 0.5-0.9m thick. Damaged by tornado 1880-1890 in Washington D.C.? There was a tornado in Washington D.C. in 1884? I recall a tornado in 1884 that damaged the Smithsonian? Let's search.

---

**Tool Call:**

"name": "web_search", "arguments": "queries": ["Washington D.C. tornado 1884 historical building damaged", "tornado 1885 Washington DC historic site", "earthquake 1897 Washington DC historic building damaged"]

---

**Tool Response:**
Tool responses are omitted

| Step 2 |
|:---:|

**Analysis:**
Seems not obvious.
Perhaps it's not US. Another capital city with tornado and earthquake: maybe Islamabad? Tornado unlikely.
Maybe it's a European capital with tornado? Tornadoes rare in Europe but can occur. Earthquake early 1900s.
Capital city beside a river: many.
Let's think of historical place like a palace or fortress.
Construction started 1830-1860, completed 1870-1880, walls 0.5-0.9m thick.
Acquired by government 1980-1990.
Damaged by tornado 1880-1890, earthquake 1890-1900.
Which country's president at acquisition time born 1920-1935? If it's a president (i.e., head of state is president), not necessarily US.
Could be a country where the capital is also named after the country? Eg. Mexico City (capital of Mexico). Tornado damage in Mexico City? Unlikely.
Maybe it's a capital in South America: e.g., Lima (Peru) beside Rímac River. Tornado damage? Unlikely.
Buenos Aires (Argentina) beside Riachuelo. Tornado? Not typical.
Brasília (Brazil) no river.
Ottawa (Canada) beside Ottawa River. Tornado in Ottawa? There was a tornado in Ottawa in 1888? Not sure.
President of Canada? Canada is a constitutional monarchy, no president.
So it's a presidential republic.
Maybe it's a small country.
Tornado damage in late 1800s could be in Bangladesh (Dhaka) but tornado rare.
Perhaps it's in Africa: capital city beside a river, tornado? Unlikely.
Could be in Central America: e.g., Guatemala City beside Paz River? Tornado?
Maybe it's in the Caribbean: e.g., Port-au-Prince (Haiti) beside Artibonite? Tornado?
Let's search for historical buildings damaged by tornado and earthquake.

---

**Tool Call:**

"name": "web_search", "arguments": "queries": ["'tornado' 1880' earthquake' 1890' historic building' capital'", "'tornado' 1885' earthquake' 1895' historical site' capital'", "'tornado' 1890' earthquake' 1897' historical building' capital'"]

---

**Tool Response:**
Tool responses are omitted

| Step 3 |
|:---:|

**Analysis:**
Maybe the tornado and earthquake are not in the same location? The place was damaged by a tornado between 1880-1890, and also suffered damage from an earthquake between 1890-1900. So two separate events damaging the same place.

Thus the place survived both.
We need to find such a place.
Let's think of famous historical places that have been damaged by tornado and earthquake.
Possibly the "Kashmir Fort" ?
Search for historical site tornado damage 1880s.

**Tool Call:**
"name": "web_search", "arguments": "queries": ["historical site tornado damage 1880s", "tornado damage to historic building 1885", "tornado 1887 historic building damage"]

**Tool Response:**
Tool responses are omitted

### Step 4

**Analysis:**
Search not effective.
Maybe the place is not well-known.
We might need to think of a specific country where tornadoes are common: USA, Bangladesh, Canada (some), Argentina (rare), etc.
If it's USA, capital Washington D.C. Historical place beside a river (Potomac). Construction started 1830-1860, completed 1870-1880. Wall thickness 0.5-0.9m.
Damaged by tornado 1880-1890 in D.C. Which building? Possibly the United States Capitol? It was damaged by tornado? Not sure.
Check if US Capitol was damaged by tornado. Search.

**Tool Call:**
"name": "web_search", "arguments": "queries": ["Washington DC historic building tornado 1884 damage", "tornado damage to Smithsonian 1885", "tornado damage to National Gallery of Art 1880s"]

**Tool Response:**
Tool responses are omitted

### Step 5

**Analysis:**
Seems not.
Maybe the historical place is not in US. Let's think of other countries where tornadoes occur: Bangladesh (tornadoes called "sudden wind"), Pakistan, India (tornadoes in some regions), Argentina (southeast), Brazil (rare), etc.
Capital city beside a river: e.g., Dhaka (Bangladesh) beside Buriganga River. Historical place built 1830-1860 start, completed 1870-1880, walls 0.5-0.9m thick. Damaged by tornado 1880-1890 in Dhaka? Not sure about tornado.
Earthquake 1890-1900 in Dhaka? There was an earthquake in 1897 (Rangpur earthquake) affecting Dhaka? Not sure.
Acquired by government 1980-1990, president at that time born 1920-1935. Bangladesh president? Bangladesh is a parliamentary republic, head of state is President (since 1971). Presidents: Sheikh Mujibur Rahman (1971-75, born 1920). He was president at acquisition? If acquisition happened 1980s, president would be not Mujib (he died 1975). So not Bangladesh.
Maybe it's Pakistan: capital Islamabad beside River? Islamabad is not on a river. Karachi is not capital. Islamabad near River Soan? Actually Islamabad is near River Soan? Not sure. Tornado in Islamabad? Unlikely.
Earthquake in Pakistan 1890s? The 1897 Quetta earthquake (1897). Not capital.
President of Pakistan in 1980s: Zia-ul-Haq (military ruler, not president). President Ghulam Ishaq Khan (1988-1993, born 1915). Not within 1920-1935.
India: capital New Delhi beside Yamuna River. Historical place built 1830-1860 start, completed 1870-1880, walls thickness 0.5-0.9m. Damaged by tornado 1880-1890 in Delhi?

Unlikely tornado in Delhi.

Earthquake 1890-1900 in Delhi? There was an earthquake in 1897? Not sure.

Acquired by government 1980-1990, president at acquisition time (President of India) born 1920-1935. Presidents of India: Zail Singh (1982-1987, born 1916). R. Venkataraman (1987-1992, born 1918). Not within 1920-1935.

So not India.

Argentina: capital Buenos Aires beside Riachuelo. Historical place built 1830-1860 start, completed 1870-1880. Damaged by tornado 1880-1890? Buenos Aires tornado rare.

Earthquake 1890-1900? Not.

President at acquisition (1980s) born 1920-1935: President Raúl Alfonsín (1983-1989, born 1927). Could be.

Historical place in Buenos Aires damaged by tornado and earthquake? Not likely.

Maybe it's a smaller country.

Consider Caribbean: Haiti capital Port-au-Prince beside Artibonite? Tornado rare.

Jamaica capital Kingston beside Kingston Harbour (not river). Tornado?

Let's think of tornadoes in Europe: tornadoes do occur, e.g., in Germany, Poland, UK.

Capital city beside a river: e.g., Warsaw (Poland) on Vistula River. Historical place built 1830-1860 start, completed 1870-1880, walls thickness 0.5-0.9m. Damaged by tornado 1880-1890 in Warsaw? Possibly a tornado hit Warsaw in 1886? Not sure.

Earthquake 1890-1900 in Warsaw? Earthquake in 1898? Not typical.

Acquired by government 1980-1990, president of Poland at that time (Poland is a republic with president) born 1920-1935. Presidents: Wojciech Jaruzelski (1989-1990, born 1923). Could be.

Historical place in Warsaw damaged by tornado and earthquake? Not aware.

Let's search for tornado in Warsaw 1880s.

---

**Tool Call:**

"name": "web_search", "arguments": "queries": ["Warsaw tornado 1885", "Warsaw tornado 1890", "Warsaw historical building tornado damage 1880s"]

---

**Tool Response:**

Tool responses are omitted

---

### Step 6

**Analysis:**

Seems not helpful.

Maybe the place is not well-known, need to think of specific historical place: a palace, fortress, monastery, etc.

Let's consider countries where president (head of state) is relevant: e.g., Mexico (president), Brazil (president), etc.

Capital city beside a river: Mexico City beside Rio Grande? Actually Mexico City is near Rio Pánuco? Not a river.

Brazil capital Brasília not on a river.

Argentina capital Buenos Aires beside Riachuelo.

Historical place built 1830-1860 start, completed 1870-1880, walls thickness 0.5-0.9m.

Damaged by tornado 1880-1890 and earthquake 1890-1900.

Acquired by government 1980-1990.

President at acquisition time born 1920-1935.

If it's Argentina, president in 1980s: Raúl Alfonsín (born 1927). So acquisition during his presidency (1983-1989). Historical place in Buenos Aires damaged by tornado and earthquake? Unlikely.

But maybe the place is in another Argentine city that is capital of a province, not national capital? The question says capital city of a country.

Let's think of African countries with tornadoes: e.g., Zimbabwe, Mozambique, South Africa (tornadoes rare). Earthquake early 1900s.

President at acquisition time born 1920-1935.

Historical place in capital city beside a river.

Maybe it's a castle in Kenya? Nairobi beside Nairobi River. Historical place built 1830s?

Not.

Tornado in Nairobi? Unlikely.

Earthquake in Kenya early 1900s? Not.

Let's consider Caribbean: Haiti capital Port-au-Prince beside Artibonite? Actually Port-au-Prince is near the Léogâne River? Not sure.

Historical place built 1830-1860 start, completed 1870-1880. Could be a fortress like Citadelle Laferrière (Haiti) built 1805-1840, not.

Tornado damage? Not.

President of Haiti in 1980s: Jean-Claude Duvalier (born 1951). Not.

Let's think of Middle East: capital city beside a river, tornado? Tornadoes in Middle East rare.

Maybe it's a place in Pakistan: capital Islamabad not on river, but Karachi not capital.

Historical place in Islamabad? Not.

Maybe it's a place in Sri Lanka: capital Colombo beside Kelani River. Historical place built 1830-1860 start, completed 1870-1880. Damaged by tornado? Unlikely.

President of Sri Lanka (head of state) in 1980s: J.R. Jayewardene (president 1978-1989, born 1906). Not.

Let's consider European countries with tornadoes: tornadoes occur in UK (e.g., Birmingham). Capital London beside Thames.

Historical place in London built 1830-1860 start, completed 1870-1880, walls thickness 0.5-0.9m. Damaged by tornado 1880-1890 in London? There was a tornado in London in 1884? I recall a tornado in 1885 near London? Not sure.

Earthquake 1890-1900 in London? Earthquake in 1896? Not typical.

Acquired by government 1980-1990, president of UK? UK is constitutional monarchy, not president.

Thus not UK.

Maybe it's a country with a president and capital city with tornado and earthquake: e.g., Philippines capital Manila beside Pasig River. Historical place built 1830-1860 start, completed 1870-1880. Damaged by tornado 1880-1890 in Manila? Tornadoes rare in Philippines (tropical cyclones). Earthquake 1897? The 1897 earthquake in Manila? Not sure.

President of Philippines in 1980s: Ferdinand Marcos (born 1917). Not within 1920-1935.

Corazon Aquino (president 1986-1992, born 1933). Yes, born 1933. Acquisition could be during her presidency (1986-1992). Historical place in Manila damaged by tornado and earthquake? Tornado in Manila? There was a tornado in Manila in 1880? Not aware.

Earthquake in Manila 1897? The 1897 earthquake in Manila? There was a major earthquake in 1897 in Philippines (Mindanao?). Not sure.

Let's search for historical building in Manila damaged by tornado 1880s.

---

**Tool Call:**

"name": "web_search", "arguments": "queries": ["Manila tornado 1885 historical building damage", "Manila earthquake 1890 historical building damage", "Manila historical site tornado 1880s earthquake 1890s"]

---

**Tool Response:**

Tool responses are omitted

---



**Step 7**



**Analysis:**

Result 1: Fort Santiago front structure destroyed by earthquake of 1880. Fort Santiago is a historical fortress in Intramuros, Manila, beside Pasig River? Intramuros is within Manila, beside Pasig River. Fort Santiago construction started? Let's check.

---

**Tool Call:**

"name": "fetch", "arguments": "url": "https://en.wikipedia.org/wiki/Fort_Santiago", "page": 1

---

**Tool Response:**

Tool responses are omitted

---

**Step 8**

**Analysis:**

Fort Santiago construction started 1590, not within 1830-1860. So not.

We need construction start 1830-1860 inclusive. So maybe a more recent building.

Historical place in capital city beside a river, damaged by tornado and earthquake in late 19th century, acquired by government 1980-1990, walls thickness 0.5-0.9m.

Let's think of a specific known building that matches tornado and earthquake damage.

Tornado damage to a historic house in Washington D.C. maybe the "Old Post Office Pavilion"? Not.

Earthquake damage in late 19th century in D.C.? The 1897 earthquake? Not.

Maybe it's not US.

Consider Canada: capital Ottawa beside Ottawa River. Historical place built 1830-1860 start, completed 1870-1880. Damaged by tornado 1880-1890 in Ottawa? There was a tornado in Ottawa in 1888? Not sure.

Earthquake in Ottawa 1890-1900? The 1896 earthquake in Ottawa? Not typical.

President of Canada? No president.

Let's think of a country with president: e.g., Indonesia (president), capital Jakarta beside Ciliwung River. Historical place built 1830-1860 start, completed 1870-1880, damaged by tornado? Tornado in Jakarta? Unlikely.

Earthquake 1890-1900 in Jakarta? The 1899 Banten earthquake? Not Jakarta.

President at acquisition time (1980s) born 1920-1935: Indonesian president Suharto (born 1921, president 1967-1998). Acquisition could be during his presidency (1980s). Historical place in Jakarta damaged by tornado? Not.

Maybe it's a palace in Thailand: capital Bangkok beside Chao Phraya River. Historical place built 1830-1860 start, completed 1870-1880. Damaged by tornado? Thailand tornado rare.

Earthquake 1890-1900 in Bangkok? Not.

President? Thailand is monarchy.

Thus not.

Maybe it's a country in South America with president: e.g., Brazil (president), capital Brasília (no river?), but Brasília not on river. Previous capital Rio de Janeiro (beside Guanabara Bay). Historical place built 1830-1860 start, completed 1870-1880, damaged by tornado? Tornado in Rio? Unlikely.

President of Brazil in 1980s: José Sarney (born 1930). He was president 1985-1990. Born 1930 within 1920-1935. So acquisition during his presidency. Historical place in Rio de Janeiro damaged by tornado and earthquake? Not.

But maybe it's a historical place in Brasília? No river.

Historical place in Rio de Janeiro beside a river (e.g., Rodrigo de Freitas Lagoon? Not river).

Maybe the "Palácio do Catete" (Presidential palace) built 1894? Not.

Let's search for historical building in Rio damaged by tornado 1880s.

---

**Tool Call:**

"name": "web_search", "arguments": "queries": ["Rio de Janeiro historical building tornado 1880s", "Rio de Janeiro earthquake 1890s historical building damage", "Rio de Janeiro palace tornado 1885"]

---

**Tool Response:**

Tool responses are omitted

---

**Step 9**

**Analysis:**

Result 3: NYTimes article about tornado at Rio in 1864. That's a tornado in 1864, outside our range (we need tornado 1880-1890). But maybe there was a tornado in Rio in 1885? Not sure.

Nevertheless, the Paço Imperial built in 18th century, not.

Let's open result 3.

---

**Tool Call:**
"name": "fetch", "arguments": "url": "https://www.nytimes.com/1864/12/02/archives/tornado-at-rio-janeiro-terrible-hail-storm-great-damage-to-the-city.html", "page": 1

**Tool Response:**
Tool responses are omitted



**Step 10**



**Analysis:**
Probably blocked.
But the tornado in Rio in 1864 is outside our window.
Thus not.
Maybe it's not Brazil.
Let's think of tornado in Europe: e.g., in Italy, tornado in 1884 in Milan? There was a tornado in Milan in 1884? Not sure.
Earthquake in Italy early 1900s: 1908 Messina earthquake. Not.
Capital city beside a river: e.g., Rome beside Tiber River. Historical place built 1830-1860 start, completed 1870-1880. Damaged by tornado? Unlikely.
President of Italy? Italy is republic with president (head of state). President in 1980s: Sandro Pertini (born 1919). Not within 1920-1935? Actually 1919 outside.
President of Italy in 1985-1992: Francesco Cossiga (born 1924). Born 1924 within range. Acquisition during his presidency (1985-1992). Historical place in Rome damaged by tornado and earthquake? Tornado in Rome? Unlikely.
Earthquake in Rome 1890s? The 1897 earthquake in Italy? Not Rome.
Maybe it's a historical place in another Italian city that is capital of region? No, capital city of country.
Let's search for historical building in Rome damaged by tornado 1880s.

**Tool Call:**
"name": "web_search", "arguments": "queries": ["Rome tornado 1880s historical building damage", "Rome earthquake 1890s historical building damage tornado", "Rome historical site tornado 1885"]

**Tool Response:**
Tool responses are omitted



**Step 11**



**Analysis:**
Search not helpful.
Maybe the tornado and earthquake are in the same country but different years.
We need to find a specific historical place, perhaps a castle or palace.
Let's think of countries where tornadoes are common and earthquakes also: e.g., USA (tornadoes common, earthquakes in some regions). So USA seems plausible.
Capital Washington D.C. Beside Potomac River.
Historical place built 1830-1860 start, completed 1870-1880. Wall thickness 0.5-0.9m ( 2-3 ft). Damaged by tornado 1880-1890 in D.C. Which tornado? There was a tornado in Washington D.C. on May 31, 1884 (the Great D.C. tornado). It damaged many buildings.
Historical place damaged by that tornado and also by an earthquake 1890-1900 (maybe the 1897 earthquake in D.C.? Actually there was an earthquake in 1897 in D.C.? Not sure.
Earthquake in 1897 in D.C.? There was an earthquake in 1897 in D.C. area? I recall the 1897 New Madrid earthquake was in 1897? Actually New Madrid earthquakes were 1811-1812. Not.
But there was an earthquake in 1898 near D.C.? Not typical.
However, there was an earthquake in 1897 in D.C. area? Let's search.

**Tool Call:**
"name": "web_search", "arguments": "queries": ["Washington DC earthquake 1890s tor-

nado 1884 historical building damaged", "Washington DC historical site tornado 1884 earthquake 1890s", "Great tornado of 1884 Washington DC damage to historical building earthquake later"]

**Tool Response:**
Tool responses are omitted

**Step 12**

**Analysis:**
Seems not straightforward.
Maybe the historical place is not a famous monument but a lesser-known house or building.
Let's consider the acquisition by government between 1980-1990. In US, many historic sites were acquired by National Park Service during that period. Example: The Old Post Office Pavilion? No.
Historical place damaged by tornado 1884 and earthquake 1897? Which building in D.C. experienced both? The United States Capitol? Not.
The White House? No tornado damage.
Perhaps it's a building in another US capital: e.g., Sacramento (California) beside American River. Historical place built 1830-1860 start, completed 1870-1880, damaged by tornado 1880-1890 in Sacramento? Unlikely.
Or state capital not national capital? The question says capital city of a country.
Maybe it's not US but another country with tornadoes: e.g., Bangladesh (tornadoes called "sudden wind") but capital Dhaka beside Buriganga River. Historical place built 1830-1860 start, completed 1870-1880, damaged by tornado 1880-1890 in Dhaka? Not sure about tornado.
Earthquake 1890-1900 in Dhaka? The 1897 Great Assam earthquake (1897) felt in Dhaka? Possibly.
Acquisition by government 1980-1990: perhaps the government of Bangladesh acquired a historical house.
President of Bangladesh at acquisition time (1980s) born 1920-1935: President Ziaur Rahman (born 1937) no. President Hussain Muhammad Ershad (born 1930). He was president 1982-1990 (military). Born 1930 within range. So acquisition during his presidency.
Historical place in Dhaka damaged by tornado and earthquake? Not sure.
Let's search for historical building in Dhaka damaged by tornado 1880s.

**Tool Call:**
"name": "web_search", "arguments": "queries": ["Dhaka tornado 1885 historical building damage", "Dhaka earthquake 1890 historical building damage", "Dhak historical site tornado 1880s earthquake 1890s"]

**Tool Response:**
Tool responses are omitted

**Step 13**

**Analysis:**
Seems not.
Maybe the country is not US or Bangladesh.
Let's think of tornadoes in Europe: e.g., in Italy, tornado in 1884 in Milan? There was a tornado in 1890 in Florence? Not sure.
Earthquake in early 1900s in same city.
Acquisition by government 1980-1990, president at acquisition born 1920-1935.
Which European country's president during 1980s had birth year within that range? For example, France: President François Mitterrand (born 1916) no. President Jacques Chirac (born 1932) president 1995-2007, not 1980s.
Germany: President (federal) Karl Carstens (born 1914) no. President Richard von Weizsäcker (born 1920). He was president 1984-1994. So acquisition could be during his presidency (1984-1994). Historical place in capital Berlin (beside Spree River). Construction started 1830-1860, completed 1870-1880, walls thickness 0.5-0.9m, damaged by

tornado 1880-1890 in Berlin? Unlikely tornado.

Earthquake 1890-1900 in Berlin? Not typical.

Maybe it's a country with president and capital city with tornado and earthquake: e.g., Mexico: President Carlos Salinas de Gortari (born 1942) no. President Miguel de la Madrid (born 1934) president 1982-1988. Born 1934 within range. Capital Mexico City beside Rio Grande? Actually Mexico City is near the River Mexico (Río Panuco? Not). Mexico City is on Valley of Mexico, no major river.

Historical place in Mexico City damaged by tornado 1880s? Unlikely.

Earthquake 1890-1900 in Mexico City: The 1896 Mexico City earthquake? Not sure.

But there was a major earthquake in 1985, not relevant.

Let's search for historical building in Mexico City damaged by tornado 1880s.

---

**Tool Call:**

"name": "web_search", "arguments": "queries": ["Mexico City tornado 1880s historical building damage", "Mexico City earthquake 1890s historical building damage tornado", "Mexico City historical site tornado 1885 earthquake 1890"]

---

**Tool Response:**

Tool responses are omitted

| Step 14 |
|:---:|

**Analysis:**

No.

Let's reconsider US.

Historical place in Washington D.C. that was damaged by tornado 1884 and earthquake 1897? Which building? The Washington Monument? It was damaged by earthquake? The Washington Monument suffered damage in 1897? I recall the 1897 earthquake in Virginia? Actually the Washington Monument was damaged in 1942? Not.

Let's search for D.C. historical site damaged by tornado 1884.

---

**Tool Call:**

"name": "web_search", "arguments": "queries": ["'tornado' 1884' Washington' historical building damaged", "'tornado' 1884' Washington DC' building damage", "'Great tornado' 1884 Washington DC historical site"]

---

**Tool Response:**

Tool responses are omitted

| Step 15 |
|:---:|

**Analysis:**

Search not picking up.

Maybe the tornado damage is not in D.C. but in another capital city where tornadoes are common: e.g., Tallahassee (Florida capital) beside a river? Tallahassee beside the Apalachicola River? Actually Tallahassee is near the Little Tallahassee River? Not major.

Historical place built 1830-1860 start, completed 1870-1880. Damaged by tornado 1880-1890 in Tallahassee? President of US at acquisition (1980-1990) born 1920-1935: President Ronald Reagan (born 1911) no, George H.W. Bush (born 1924). Acquisition of a historical place in Tallahassee by US government? Not.

But the question says "the president of the country at the time of its acquisition was born between 1920 and 1935 (inclusive)." So the country's president, not necessarily US.

If it's a US state capital, the country is the US, president is US president.

Thus it's a US capital city (Washington D.C.) historical place.

Thus we need a historical place in Washington D.C. that was damaged by tornado 1884 and earthquake 1897 (maybe the 1897 earthquake in D.C. area? Actually there was an earthquake in 1897 in D.C.? I recall the 1897 "New Madrid" earthquake was 1811-1812. There was a minor earthquake in 1897 in D.C.? Not sure.

Let's search for earthquake damage to Washington D.C. historical sites in late 19th century.

---

**Tool Call:**
"name": "web_search", "arguments": "queries": ["Washington DC earthquake 1890s historical building damage", "Washington DC earthquake 1897 building damage", "Washington DC earthquake 1895 historical site"]

---

**Tool Response:**
Tool responses are omitted

---

### Step 16 to Step 49

**(The medium steps have been omitted.)**

---

### Step 50

**Analysis:**
Scrolling for info about tornado damage.

---

**Tool Call:**
"name": "find", "arguments": "url": "https://en.wikipedia.org/wiki/Ruplal_House", "query": "tornado"

---

**Tool Response:**
Tool responses are omitted

---

### Step 51

**Analysis:**
The Wikipedia page doesn't mention tornado. But earlier search result indicated that Ahsan Manzil had tornado damage. Let's open Ahsan Manzil page.

---

**Tool Call:**
"name": "fetch", "arguments": "url": "https://en.wikipedia.org/wiki/Ahsan_Manzil", "page": 1

---

**Tool Response:**
Tool responses are omitted

---

### Step 52

**Analysis:**
Great! Ahsan Manzil matches many criteria:
- Capital city of a country: Dhaka, capital of Bangladesh.
- Beside a river: Buriganga River.
- Construction started 1859 (within 1830-1860), completed 1872 (within 1870-1880).
- Wall thickness: Need to verify 0.5-0.9 meters. Need evidence. Let's find wall thickness info.

---

**Tool Call:**
"name": "find", "arguments": "url": "https://en.wikipedia.org/wiki/Ahsan_Manzil", "query": "wall thickness"

---

**Tool Response:**
Tool responses are omitted

---

### Step 53

**Analysis:**
Search for wall thickness of Ahsan Manzil.

---

**Tool Call:**
"name": "web_search", "arguments": "queries": ["Ahsan Manzil wall thickness", "Ahsan Manzil walls thickness meters", "Ahsan Manzil construction walls thickness"]

**Tool Response:**
Tool responses are omitted

### Step 54

**Analysis:**
Thus wall thickness 0.78 m fits criteria 5.
Now we have evidence for tornado damage (April 7 1888 tornado) and earthquake repair after 1897 Assam earthquake (criterion 7 and 8). Also acquisition by government in 1985 (criterion 6). President of Bangladesh at acquisition time (1985) was Hussain Muhammad Ershad (born 1930, within 1920-1935). So all criteria satisfied.
Thus answer: Ahsan Manzil.
Provide answer with citations.
We need to cite sources for each claim.
We have Wikipedia page for Ahsan Manzil with tornado and earthquake info. Let's capture relevant lines.
Find tornado mention.

**Tool Call:**
"name": "find", "arguments": "url": "https://en.wikipedia.org/wiki/Ahsan_Manzil", "query": "tornado"

**Tool Response:**
Tool responses are omitted

### Step 55

**Analysis:**
Find earthquake mention.

**Tool Call:**
"name": "find", "arguments": "url": "https://en.wikipedia.org/wiki/Ahsan_Manzil", "query": "earthquake"

**Tool Response:**
Tool responses are omitted

### Step 56

**Analysis:**
Scrolling near that area for earthquake mention.

**Tool Call:**
"name": "fetch", "arguments": "url": "https://en.wikipedia.org/wiki/Ahsan_Manzil", "page": 1

**Tool Response:**
Tool responses are omitted

### Step 57

**Analysis:**
We have evidence for tornado and earthquake.
Need evidence for wall thickness 0.5-0.9 m. Search within page for "thickness".

**Tool Call:**
"name": "find", "arguments": "url": "https://en.wikipedia.org/wiki/Ahsan_Manzil", "query": "thickness"

---

**Tool Response:**
Tool responses are omitted

---

### Step 58

**Analysis:**
Maybe wall thickness info not in Wikipedia. Need another source.
Search for Ahsan Manzil wall thickness.

---

**Tool Call:**
"name": "web_search", "arguments": "queries": ["'Ahsan Manzil' wall thickness", "'Ahsan Manzil' walls thickness", "'Ahsan Manzil' brick thickness"]

---

**Tool Response:**
Tool responses are omitted

---

### Step 59

**Analysis:**
Thus we have multiple sources for wall thickness.
We can use Banglapedia entry.
Let's open Banglapedia page.

---

**Tool Call:**
"name": "fetch", "arguments": "url": "https://en.banglapedia.org/index.php/Ahsan_Manzil", "page": 1

---

**Tool Response:**
Tool responses are omitted

---

### Step 60

**Analysis:**
We have evidence for:
- Construction start 1859, completed 1872.
- Tornado damage on April 7, 1888.
- Earthquake damage on 12 June 1897.
- Wall thickness 0.78 m.
- Acquired by government in 1985.
- President at acquisition time: Hussain Muhammad Ershad (born 1930).
Now we need to cite president info.
Search for president of Bangladesh in 1985.

---

**Tool Call:**
"name": "web_search", "arguments": "queries": ["Bangladesh president 1985 Ershad born 1930", "Hussain Muhammad Ershad president 1985 birth 1930", "Bangladesh president 1985 born 1930"]

---

**Tool Response:**
Tool responses are omitted

---

### Step 61

**Analysis:**
Thus we have sources.
Now we need to answer: name of historical place: Ahsan Manzil.
Provide answer with citations.
We'll cite Banglapedia for construction dates, tornado, earthquake, wall thickness, acquisition. Also cite Wikipedia for president birth and presidency.
Let's open Wikipedia page for Hussain Muhammad Ershad.

---

**Tool Call:**
"name": "fetch", "arguments": "url": "https://en.wikipedia.org/wiki/Hussain_Muhammad _Ershad", "page": 1

---

**Tool Response:**
Tool responses are omitted

**Final Response**

We have evidence for president birth 1930 and presidency 1983-1990.
Now we have all needed evidence.
Now craft final answer.
<|answer_start|>
The historical place is **Ahsan Manzil**, the former palace of the Nawabs of Dhaka in Bangladesh.
**Why Ahsan Manzil fits all the criteria**

| Criterion | Evidence |
|———-|———-|
| **Capital city of a country** | Ahsan Manzil is located in Dhaka, the capital of Bangladesh. |
| **Situated beside a river** | It stands on the bank of the Buriganga River. |
| **Construction started between 1830-1860** | Construction of the palace began in **1859**. |
| **Construction completed between 1870-1880** | The palace was completed in **1872**. |
| **Wall thickness 0.5-0.9 m** | The walls of the palace are **about 0.78 m thick**. |
| **Acquired by the government between 1980-1990** | The government of Bangladesh **acquired Ahsan Manzil and its surroundings in 1985**. |
| **Damaged by a tornado between 1880-1890** | The palace was **partly damaged by a tornado on 7 April 1888**. |
| **Damaged by an earthquake between 1890-1900** | It was **badly damaged by the earthquake of 12 June 1897**. |
| **President of the country at the time of acquisition was born between 1920-1935** | The president of Bangladesh in 1985 was **Hussain Muhammad Ershad**, who was **born in 1930** (within the required range). |

All of these details are documented in the Banglapedia entry for Ahsan Manzil and the Wikipedia page for Hussain Muhammad Ershad. Hence, the answer is **Ahsan Manzil**.
<|answer_end|>

