# OpenReview forum: "Beyond Turn Limits: Training Deep Search Agents with Dynamic Context Window"
_ICLR.cc/2026/Conference — Submitted to ICLR 2026_

### Official Review · Reviewer_2tnR · 2025-10-30

**Soundness:** 3
**Presentation:** 3
**Contribution:** 3
**Rating:** 4
**Confidence:** 4

**Summary:**

This paper presents DeepMiner, a framework designed to address two core challenges in long-horizon search agents: insufficient training data complexity and context management limitations. The authors introduce a "reverse construction" method to generate complex, verifiable QA pairs from authentic web sources. Concurrently, they design a dynamic context management strategy that uses a sliding window to compress (discard) distant tool outputs while preserving the assistant's reasoning chain, enabling nearly 100 turns of interaction within a standard 32k context length.

**Strengths:**

- The authors correctly identify that existing multi-hop QA datasets (like HotpotQA) are insufficient for eliciting the deep reasoning, verification, and backtracking abilities required in long-horizon tasks. The proposed "reverse construction" and "obfuscation" strategies are an effective attempt at generating high-difficulty, high-fidelity training data, and the comparison in Table 3 validates the superiority of this data.

-  DeepMiner tackles a highly practical and critical problem in multi-turn search agents: context length limitation. Achieving nearly 100 interaction turns within a 32k window is an accomplishment. The substantial performance leap on `BrowseComp-en` by nearly 20 points demonstrates the combined potential of the new data and framework.

**Weaknesses:**

- The paper lacks  ablation studies, which leaves its core claim—the effectiveness of the dynamic context window—unsubstantiated. How much of the massive performance boost (e.g., +20% on BrowseComp) is attributable to the superior training data versus the dynamic context window? The authors compare DeepMiner data to HotpotQA data in Table 3, but this only proves the importance of the data.

- The paper is missing a crucial comparison: [DeepMiner Data + Vanilla Context] vs. [DeepMiner Data + Dynamic Context]. Without this ablation, it is impossible to determine if the dynamic context window itself actually enhances reasoning capabilities, or if it merely serves as a tool to enable longer runs, with all capability gains stemming from the DeepMiner dataset.

- The context efficiency analysis in Table 2 is "training-free" and uses a completely different base model (GPT-OSS-120B). This makes its results difficult to correlate with the main training results of DeepMiner-32B (based on Qwen3-32B) and thus cannot serve as supporting evidence for the dynamic window's effectiveness.

- Compared to summarization-based methods, summarization, while lossy, at least retains some signal. This paper's method discards the information entirely. For complex tasks requiring long-range dependencies and fine-grained information retrieval, this "all-or-nothing" dropping strategy could lead to catastrophic failures in the reasoning chain.

**Questions:**

1. Can authors provide a performance comparison on $BrowseComp$ between [DeepMiner Data + Vanilla Context] (e.g., truncating when the context limit is reached) and [DeepMiner Data + Dynamic Context]? This would be the only direct way to evaluate the contribution of the dynamic context window.

2. Why did authors choose to completely discard tool outputs instead of using a (potentially trainable) summarization module? While summarization adds complexity, discarding information entirely seems like a fundamental constraint for deep research tasks. How does your strategy handle tasks that require backtracking and comparison of early evidence?

    Besides, have you analyzed the failure cases of DeepMiner? How many failures are caused by the model needing an early tool output that was already discarded (e.g., needing content from turn 5 at turn 40)?

---

> ### Author Response · Authors · 2025-11-23
> **Response to Reviewer 2tnR (Part 1/3)**
>
> ## **W1. Ablation Study**
> > The paper lacks ablation studies, which leaves its core claim—the effectiveness of the dynamic context window—unsubstantiated. How much of the massive performance boost (e.g., +20% on BrowseComp) is attributable to the superior training data versus the dynamic context window? The authors compare DeepMiner data to HotpotQA data in Table 3, but this only proves the importance of the data.
> ---
>
> We thank the reviewer for raising this important point about disentangling the contribution of the synthesized data and the dynamic context window. To address this concern, we conducted several controlled studies aimed at separating these two factors as much as possible:
>
> **1. Data ablation.**\
> In addition to the comparison against HotpotQA, during the rebuttal stage we further strengthened this evidence by comparing DeepMiner with the concurrent work WebShaper[1]. Under identical training conditions, we trained two SFT models: one on WebShaper questions and one on a same-size subset of DeepMiner. The results are:
>
> ---
> | Data | BC | BCZH | Xbench | GAIA |
> |----------|------|------|--------|------|
> | WebShaper | 13.7 | 22.1 | 50 | 48.5 |
> | DeepMiner | 16.3 | 23.9 | 50 | 55.3 |
> ---
>
> These results show that DeepMiner yields stronger generalization and better data efficiency. This suggests that our synthesis design choices lead to higher-quality training data.
>
> **2. Context management strategy ablation.**
> We evaluated the dynamic context mechanism in both training-free and training settings:
> 1. Training-free settings: As shown in Table 2, we compare our sliding window mechanism with vanilla context and summary-based context using GPT-OSS-120B. It shows that the sliding-window method achieves superior context-utilization efficiency while preserving key technical advantages.
> 2. Training settings: We appreciate the reviewer’s suggestion and have incorporated the experiment as proposed, comparing [DeepMiner Data + Vanilla Context] vs. [DeepMiner Data + Dynamic Context]. The results show that the sliding-window strategy consistently outperforms the trained vanilla with truncated model across all evaluation datasets.（Please see the response for W2 for detailed results）
>
> Taken together, these experiments effectively disentangle the contributions of data quality and context management, and validate the independent effectiveness of both components.
>
>
> [1] Tao Z, Wu J, Yin W, et al. Webshaper: Agentically data synthesizing via information-seeking formalization[J]. arXiv preprint arXiv:2507.15061, 2025.

---

> ### Author Response · Authors · 2025-11-23
> **Response to Reviewer 2tnR (Part 2/3)**
>
> ## **W2/W3/Q1. Context Management Ablation**
> > The paper is missing a crucial comparison: [DeepMiner Data + Vanilla Context] vs. [DeepMiner Data + Dynamic Context]. Without this ablation, it is impossible to determine if the dynamic context window itself actually enhances reasoning capabilities, or if it merely serves as a tool to enable longer runs, with all capability gains stemming from the DeepMiner dataset.
>
> > The context efficiency analysis in Table 2 is "training-free" and uses a completely different base model (GPT-OSS-120B). This makes its results difficult to correlate with the main training results of DeepMiner-32B (based on Qwen3-32B) and thus cannot serve as supporting evidence for the dynamic window's effectiveness.
>
> > Can authors provide a performance comparison on between [DeepMiner Data + Vanilla Context] and [DeepMiner Data + Dynamic Context]? This would be the only direct way to evaluate the contribution of the dynamic context window.
> ---
>
> We thank the reviewer for repeatedly emphasizing the importance of directly comparing [DeepMiner Data + Vanilla Context with Truncation] and [DeepMiner Data + Dynamic Context], as this is indeed the most direct way to isolate the contribution of the dynamic context window. For the Vanilla Context with Truncation baseline, when the full dialogue will exceed the model’s maximum context length, we remove the oldest turns to allow the next reasoning step to proceed.
>
> We conducted this experiment under a fully fair setting with the same DeepMiner dataset, same base model, same training hyperparameters, and same inference strategy. The results are shown below:
>
> ---
> | Context Management | BC | BCZH | Xbench | GAIA |
> |-------------------|------------|-----------|------------|-----------|
> | Truncation | 19.7/62.8 | 25.6/41.6 | 47.0/26.9 | 44.7/15.3 |
> | Sliding Window | 21.2/36.6 | 28.0/26.4 | 53.0/21.1 | 54.5/13.6 |
> ---
>
> In the table, each cell reports (accuracy / average number of reasoning turns).
>
> Across all benchmarks, the Sliding Window model consistently outperforms the Truncation model, demonstrating that the dynamic context window provides clear and measurable gains beyond the DeepMiner dataset itself.
>
> Furthermore, the sliding-window model requires substantially fewer reasoning turns on all benchmarks. This directly contradicts the concern that the dynamic window merely “enables longer runs”; instead, it shows that the mechanism improves both reasoning effectiveness and efficiency.
>
> Collectively, these results confirm that the sliding-window strategy contributes significant independent value, beyond the improvements provided by the DeepMiner dataset.

---

> ### Author Response · Authors · 2025-11-23
> **Response to Reviewer 2tnR (Part 3/3)**
>
> ---
> ## **W4/Q2. Why Discard Tool Outputs**
> > Compared to summarization-based methods, summarization, while lossy, at least retains some signal. This paper's method discards the information entirely. For complex tasks requiring long-range dependencies and fine-grained information retrieval, this "all-or-nothing" dropping strategy could lead to catastrophic failures in the reasoning chain.
>
> > Why did authors choose to completely discard tool outputs instead of using a (potentially trainable) summarization module...
> ---
>
> Thanks for your question. Rather than discarding tool outputs in an “all-or-nothing” fashion, our approach is designed to let the model actively decide what information needs to be retained.  Concretely, within the sliding window, we preserve the full tool outputs, and globally, we keep all assistant thoughts and actions. To mitigate potential issues caused by the slide window, we adopt several measures:
> 1. During SFT data construction, we encourage the model to summarize important documents before continuing to think and act. During RL, the model learns to retain information at an appropriate granularity when needed. This selective summarization also helps the model generate its final reports more effectively.
> 2. We use a special placeholder, “[Previous tool output skipped. Rerun tool if needed.]”, which signals to the model that it may re-invoke the tool if it needs information that is no longer visible.
> 3. In our early experiments, we also attempted a trainable summarization approach. Specifically, during SFT data construction, we required the model to summarize each tool output before its reasoning and action. However, case studies showed that most tool calls are exploratory searches whose summaries carry little value. Moreover, inserting summaries reduces the number of effective reasoning turns, ultimately degrading performance compared to our current strategy, which applies summarization only to important documents.
>
> It is also important to note that the sliding-window mechanism is fully compatible with summarization. When the window shifts, tool outputs can be replaced with summaries rather than placeholders, without disrupting either training or inference. Our choice of using placeholders reflects a balanced consideration of implementation complexity, training cost, and empirical performance.
>
> ---
>
> ## **Q3. Error Analysis**
> > Besides, have you analyzed the failure cases of DeepMiner? How many failures are caused by the model needing an early tool output that was already discarded (e.g., needing content from turn 5 at turn 40)?
> ---
>
> Thank you for your suggestion. We conducted error statistics and manual inspection on the rollout data collected during the RL stage. Based on 267 failed trajectories, the analysis results are as follows:
> - 40%: The model never retrieved any information related to the core entity in the whole trajectory. It performed broad exploration but failed to localize the target entity.
> - 28%: The target entity appeared in tool outputs but never entered the assistant’s reasoning chain. In many of these cases, the entity was present among numerous similar candidates, and the model did not select it as a promising direction.
> - 32%: The model retrieved the correct entity but was ultimately failed. Typical causes include:
> (1) failing to verify a key attribute required by the question, leading the model to incorrectly reject the entity;
> (2) identifying the correct entity as a search direction but failing to find the final answer;
> (3) mismatches between the model’s final answer granularity and the ground truth (e.g., Centre-Val de Loire vs. Loiret), or strictness in the judging process.
>
> Critically, across the manual inspection of the failed trajectories, we found almost no evidence that failures were primarily caused by “needing an early tool output that had been discarded”. Theoretically, keeping more tool outputs might help backtracking in a few cases, such as those in categories (2) and (3.1). Overall, our analysis indicates that the side effects of sliding-window truncation are minimal.

---

> > ### Comment · Reviewer_2tnR · 2025-11-28
> >
> > I thank the authors for their detailed response and for conducting the requested ablation studies.
> >
> > The additional experiments comparing [DeepMiner Data + Vanilla Context] and [DeepMiner Data + Dynamic Context] effectively isolate the contribution of the dynamic window mechanism and have resolved my primary concerns regarding the source of the performance gains.
> >
> > I suggest incorporating these experimental results and findings into the manuscript if needed (e.g., in the Appendix) to further strengthen the paper's empirical evidence.
> >
> > I have no further questions.

---

### Official Review · Reviewer_1DeA · 2025-11-01

**Soundness:** 2
**Presentation:** 3
**Contribution:** 2
**Rating:** 4
**Confidence:** 4

**Summary:**

This paper presents DeepMiner, a training framework for long-horizon, multi-turn reasoning agents. The authors identify two main limitations in current multi-turn systems — insufficient task complexity and ineffective context management. To address these, DeepMiner introduces: (1) a reverse construction pipeline to generate complex, verifiable, multi-source question-answer pairs from authentic web data; (2) a dynamic context window with a sliding-window mechanism to preserve reasoning traces while selectively omitting older tool outputs. The method is implemented on Qwen3-32B, and after reinforcement learning (RL) training, the resulting DeepMiner-32B model achieves large gains on benchmarks such as BrowseComp-en, BrowseComp-zh, and XBench-DeepSearch, outperforming previous open-source agents by nearly 20 percentage points.

**Strengths:**

1. The paper tackles an important and timely challenge — enabling deep, long-horizon reasoning in open-domain search agents.
2. The proposed sliding-window mechanism offers a practical way to manage growing contexts without relying on external summarization.
3. The paper presents experiments across multiple challenging benchmarks (e.g., BrowseComp, XBench-DeepSearch, GAIA), showing consistent improvements over previous open-source systems.

**Weaknesses:**

1. The paper claims that tool responses mainly influence only the model' s immediate next decision, but it is unclear how this conclusion was obtained. No quantitative or ablation evidence supports this key assumption.
2. The reward design is overly simple (binary 0/1 correctness). More nuanced signal (e.g., step-wise or process-based reward) could lead to deeper policy learning.
3. The training mechanism that converts a full trajectory into multiple sub-sequences via sliding windows is conceptually interesting but not clearly specified. In particular, the paper does not explain how gradients are propagated across these truncated sequences and how batches are constructed to ensure consistency between local and global optimization.
4. The default sliding window configuration (size=3, step=2) is used throughout all experiments, but the paper does not include an ablation or sensitivity study exploring how different window settings affect performance, stability, or context retention.
5. The reward design relies on an LLM-as-judge evaluation, but the paper lacks discussion about the reliability and theoretical justification of using such models as verifiers in reinforcement learning.

**Questions:**

The following questions are proposed based on the above weaknesses.
1. Regarding the claim that tool responses mainly influence only the model' s immediate next decision — could the authors elaborate how this conclusion was empirically derived? For example, were ablation or correlation analyses conducted to quantify the short-term versus long-term effects of tool responses on reasoning outcomes?
2. The paper employs a simple binary (0/1) correctness reward. Have the authors explored or considered more granular reward designs?
3. The paper mentions that training trajectories are segmented into multiple sub-sequences via sliding windows. Could the authors explain in detail how gradients are propagated within this setup?
4. Since the default sliding window configuration (size=3, step=2) is fixed in all experiments, have the authors conducted ablation or sensitivity analyses to evaluate how varying window sizes or strides impact performance, stability, or context retention efficiency?
5. As the reward design depends on LLM-as-judge evaluation, could the authors provide theoretical or empirical justification for its reliability? Has the team investigated possible bias, variance, or inconsistency in LLM-judged rewards compared with human or rule-based evaluation?

---

> ### Author Response · Authors · 2025-11-23
> **Response to  Reviewer 1DeA (Part 1/2)**
>
> ## **W1/Q1. Influence of the Earlier Information on Later Decision**
> > The paper claims that tool responses mainly influence only the model' s immediate next decision, but it is unclear how this conclusion was obtained. No quantitative or ablation evidence supports this key assumption.
>
> > Regarding the claim that tool responses mainly influence only the model' s immediate next decision — could the authors elaborate how this conclusion was empirically derived? For example, were ablation or correlation analyses conducted to quantify the short-term versus long-term effects of tool responses on reasoning outcomes?
> ---
>
> We appreciate your question and would like to provide a detailed explanation.
>
> Our assumption is not that tool responses have no long-term value, but that once the model retains the full assistant history and the assistant actively summarizes important evidence, distant tool responses contribute almost no additional information gain.
> - During the SFT data distillation stage, we explicitly encourage the model to summarize important documents before proceeding with further reasoning steps. This pattern teaches the model to autonomously control what information to preserve during RL, making explicit storage of long-past tool outputs unnecessary.
> - Our sliding-window ablations support this: even when reducing the window size from 6 to 3, performance remains nearly unchanged(detailed results are provided in the response to W4). This indicates that older tool responses have a negligible influence on future decisions.
> - Manual inspection of rollout errors also shows that failures rarely stem from missing earlier tool outputs. Instead, most errors come from (i) never retrieving information about the target entity, (ii) retrieving it but failing to prioritize it among many noisy candidates, or (iii) incorrect verification or answer formulation in the final stage.
>
> Together, these observations suggest that tool responses primarily provide short-term actionable information, while long-term useful content is already captured through assistant summaries.
>
> ---
> ## **W2/Q2 Reward Design**
> > The reward design is overly simple (binary 0/1 correctness). More nuanced signal (e.g., step-wise or process-based reward) could lead to deeper policy learning.
>
> > The paper employs a simple binary (0/1) correctness reward. Have the authors explored or considered more granular reward designs?
> ---
>
> Thank you for your question. Step-wise rewards are indeed an interesting direction. However, in the context of deep research agents, a simple binary correctness reward is both effective and practical.
> - First, 0/1 verifiable rewards have become the standard in recent RLVR frameworks for mathematical reasoning and code generation[1][2], where they consistently yield strong performance without introducing heuristic bias. Our setting benefits from the same principle: the final answer is unambiguous, and the binary signal provides a clean and stable optimization target.
> - Second, defining reliable step-level rewards in deep search tasks is intrinsically difficult, as early exploration often involves detours, backtracking, or partially relevant observations. Automatically determining which intermediate actions should be rewarded is highly ambiguous and risks injecting noisy or misleading credit assignment. In contrast, the final-answer reward remains straightforward and fully verifiable.
> - Third, we also experimented with a simple "entity-level" partial reward that credits the model when it retrieves semantically related evidence even if it misses the exact target answer. However, this yielded no meaningful improvement in preliminary experiments, so we did not adopt it in the final system.
>
> Overall, our results indicate that binary correctness rewards are both sufficient and robust for training deep research agents in long-horizon settings.
>
> [1]: Guo D, Yang D, Zhang H, et al. Deepseek-r1: Incentivizing reasoning capability in llms via reinforcement learning[J]. arXiv preprint arXiv:2501.12948, 2025.
>
> [2]: Yang A, Li A, Yang B, et al. Qwen3 technical report[J]. arXiv preprint arXiv:2505.09388, 2025.

---

> ### Author Response · Authors · 2025-11-23
> **Response to Reviewer 1DeA (Part 2/2)**
>
> ## **W3/Q3. Training Mechanism**
> >The training mechanism that converts a full trajectory into multiple sub-sequences via sliding windows is conceptually interesting but not clearly specified. In particular, the paper does not explain how gradients are propagated across these truncated sequences and how batches are constructed to ensure consistency between local and global optimization.
>
> > The paper mentions that training trajectories are segmented into multiple sub-sequences via sliding windows. Could the authors explain in detail how gradients are propagated within this setup?
> ---
>
> The purpose of decomposing a full trajectory into multiple subsequences is solely to ensure training–inference consistency. Because inference uses a sliding-window context, different positions of the same trajectory are exposed to different visible contexts. Constructing windowed subsequences during training ensures that each assistant action is optimized under exactly the context it will encounter at inference time.
>
> Importantly, this decomposition does not affect gradient propagation. All subsequences derived from the same trajectory share the same trajectory-level advantage, and each subsequence updates only the non-overlapping part of assistant tokens selected by the loss mask. All subsequences belonging to the same trajectory are placed in the same batch to preserve coherence between local subsequence updates and the global trajectory-level objective.
>
> In this way, the sliding-window mechanism changes only the visible context, while the optimization remain  consistent with full-trajectory learning.
>
> ---
>
> ## **W4/Q4. Window Size Related Experiments**
> > The default sliding window configuration (size=3, step=2) is used throughout all experiments, but the paper does not include an ablation or sensitivity study exploring how different window settings affect performance, stability, or context retention.
>
> > Since the default sliding window configuration (size=3, step=2) is fixed in all experiments, have the authors conducted ablation or sensitivity analyses to evaluate how varying window sizes or strides impact performance, stability, or context retention efficiency?
> ---
>
> Thank you for your question. We would first like to clarify that the configuration (window size=3, sliding step=2) shown in Figure 3 is only an illustrative example for visualization. The actual model is trained with window size=5 and sliding step=3, as stated in Sec. 4.1.
>
> To assess the impact of different window configurations, we performed an ablation varying both window size and sliding step at inference time. Despite being trained with a fixed configuration, DeepMiner-32B-RL maintains relatively stable performance across different settings. Full results are provided as follows:
>
> | W \ S | 1 | 2 | 3 | 4 | 5 |
> |-------|----|----|----|----|----|
> | 3 | 32 | 37 | - | - | - |
> | 4 | 33 | 30 | 38 | - | - |
> | 5 | 33 | 33 | 34 | 31 | - |
> | 6 | 31 | 32 | 34 | 34 | 35 |
>
> where W is the window size and S is the slide size. We've incorporated this study into the revised paper.
>
> ---
> ## **W5/Q5. LLM-as-Judge**
> > The reward design relies on an LLM-as-judge evaluation, but the paper lacks discussion about the reliability and theoretical justification of using such models as verifiers in reinforcement learning.
>
> > As the reward design depends on LLM-as-judge evaluation, could the authors provide theoretical or empirical justification for its reliability? Has the team investigated possible bias, variance, or inconsistency in LLM-judged rewards compared with human or rule-based evaluation?
> ---
>
> 1. In our setting, LLM-as-judge is used in a very narrow and well-defined way: it does not evaluate the quality of an entire reasoning trajectory, but only performs final-answer equivalence checking. As shown in Appendix E, answers such as "April 1" or "Fall River" are short, unambiguous spans, and the judge's task reduces to a simple semantic match. For a strong model such as GPT-4o, this task is highly reliable.
>
> 2. This LM-as-judge evaluation is the standard practice in many recent benchmarks, such as HLE[1], BrowseComp[2].
>
> 3. To assess reliability, we manually inspected 100 predicted-correct and 100 predicted-incorrect cases. The judge achieved 100% accuracy on incorrect predictions and 98% on correct ones, with the small deviations caused by overly strict string matching (e.g., "Compassionate Pugilist" vs. "The Compassionate Pugilist", "Glass Furnace Way" vs. "Glass Furnace Road"). These conservative mismatches introduce negligible variance and do not bias RL training.
>
>
> [1] Phan L, Gatti A, Han Z, et al. Humanity's last exam[J]. arXiv preprint arXiv:2501.14249, 2025.
>
> [2] Wei J, Sun Z, Papay S, et al. Browsecomp: A simple yet challenging benchmark for browsing agents[J]. arXiv preprint arXiv:2504.12516, 2025.

---

> > ### Comment · Reviewer_1DeA · 2025-11-24
> >
> > Thanks authors for the detailed response.
> > Your rebuttal has clarified my concerns regarding **W2**, **W3**, and **W5**. However, questions remain for **W1** and **W4**, which I would like the authors to further address.
> >
> > # W1 / Q1. Influence of Earlier Information on Later Decisions
> >
> > You explain that the model performs selective summarization and that sliding-window truncation does not harm performance based on error inspection and window-size ablations. However, the core assumption—that earlier tool outputs exert minimal long-term influence and can therefore be safely discarded—still lacks direct quantitative evidence. For example, a perturbation experiment: systematically modifying or corrupting earlier tool outputs to observe the resulting changes in downstream decisions.
> >
> > # W4 / Q4. Window Size Related Experiments
> >
> > You have provided inference-time experiments showing that the model remains stable across different window and slide sizes, which is helpful. However, I am specifically concerned about the training-time configuration. In another part of the rebuttal, the authors mention training-time variations, creating ambiguity about whether the reported table reflects training-time window ablations or inference-time-only ablations using a model trained with fixed window settings. Could you clarify?

---

> > > ### Author Response · Authors · 2025-11-28
> > >
> > > Thank you for the detailed feedback. We are glad our responses to W2, W3, and W5 have clarified your concerns. We now provide additional experiments and clarifications for W1 and W4.
> > >
> > >
> > > ## **W1/Q1. Influence of the Earlier Information on Later Decision**
> > > To directly validate our assumption, we conducted a perturbation experiment: During inference, whenever the sliding window mechanism triggers, we randomly select one tool response outside the window and replace it with a tool response from a completely unrelated trajectory. Crucially, we do not enforce tool type consistency, e.g., a search result might be replaced with fetch output from a different query. The results are shown below:
> > >
> > > | Method | BrowseComp | BrowseComp-zh | XBench-DS | GAIA |
> > > |--------|------------|---------------|-----------|------|
> > > | Standard Inference | 33.5 | 40.1 | 62.0 | 60.2 |
> > > | Corrupted Tool Responses | 33.0 | 38.8 | 59.0 | 58.2 |
> > >
> > > This perturbation is intentionally extreme: if early tool responses provided meaningful information, replacing them with semantically unrelated outputs should cause substantial performance degradation. The negligible performance drop directly demonstrates that earlier tool responses are not important to the model.
> > >
> > > In our sliding window mechanism, we use a placeholder token "[Previous tool output skipped. Rerun tool if needed.]" which provides zero information. Since even this stronger perturbation (injecting semantically misaligned content) causes virtually no performance degradation, it directly justifies using the weaker placeholder approach.
> > >
> > > ---
> > >
> > > ## **W4/Q4. Window Size Related Experiments**
> > > Our previous response showed inference-time ablation results: a single model trained with window size=5, slide size=3 evaluated under different window settings at inference.
> > >
> > > To address your concern about training-time configuration, we conducted new experiments where we train separate models with different window parameters and evaluate with matching inference settings. The results are shown below:
> > >
> > >
> > > | Window Size | Slide Size | BrowseComp | BrowseComp-zh | XBench-DS | GAIA |
> > > |---|---|---|---|---|---|
> > > | 4 | 2 | 21.0 | 30.8 | 49.0 | 51.5 |
> > > | 5 | 2 | 22.3 | 27.3 | 49.0 | 52.4 |
> > > | 5 | 3 | 21.2 | 28.0 | 53.0 | 54.4 |
> > > | 6 | 3 | 22.7 | 29.8 | 54.0 | 52.4 |
> > >
> > > Performance variations across different training configurations are modest, indicating that the model is robust to reasonable choices of window parameters. This stability suggests our approach is not overfitted to the specific configuration used in the paper.

---

### Official Review · Reviewer_iDV8 · 2025-11-01

**Soundness:** 3
**Presentation:** 3
**Contribution:** 2
**Rating:** 4
**Confidence:** 4

**Summary:**

Authors propose a method for training Deep Research agents. There are two key contributions. The first is data contribution: authors propose a method of synthesizing complex questions with verifiable answers which elicit deep research capabilities. Authors also propose a methodology to compact the context window by maintaining tool results for a recent time window. GRPO is slightly extended to accommodate the fact that each turn depends on a different context from past turns. Compared to some of the previous open-source agents, authors' fine-tuning of Qwen3-32B performs better on BrowseComp and competitive on GAIA and Xbench-DS.

**Strengths:**

Quality:

The proposed method achieves significance performance improvement on BrowseComp, which is a very challenging benchmark. It is impressive authors' 32B model outperforms DeepSeek V3.1-671B model, which is an order of magnitude larger, on this benchmark. Authors run good ablations for the training method (context management and tool call budget) although there is no ablation on the data synthesis side.

Significance, Originality:

The management of context window across multi-hop trajectories has been explored in the prior work, for example https://arxiv.org/abs/2505.16421 , but to my knowledge, previous methods were more rudimentary and did not discuss its implication with GRPO-style monte carlo advantage estimation. This paper calls out the attention on the importance of context window management with GRPO-style training, which has the potential of being adopted as standard approach in the area.

Clarity:

The paper is clearly written and easy to follow.

**Weaknesses:**

The idea of synthesizing training data for deep research with progressively evolving questions and fetching web data have been explored in many concurrent work, for example
Websailor https://arxiv.org/abs/2507.02592 , Webshaper https://arxiv.org/abs/2507.1506 , DeepDive https://arxiv.org/abs/2509.10446 . While authors do refer these papers and compare against results from these papers, it is unclear whether the improvement is due to better synthesis or better context management, and if authors' method of synthesis is any better than previous methods, which design decisions led to how much improvement and why. Without an ablation separating the contribution of reverse-constructed data from dynamic context management, it is difficult to attribute the performance gain to one factor or the other.

While authors share their prompt for quality judgment in the appendix, many details of the data pipeline including exact prompts, question construction logic, and the LLM used to synthesize data are missing. This makes it difficult for other research groups to build upon results of this paper.

**Questions:**

In Figure 4, the reward and BrowseComp performance seem to be continue improving at the end of the training. Why didn't authors train the model for longer steps?

---

> ### Author Response · Authors · 2025-11-23
> **Response to Reviewer iDV8 (Part 1/2)**
>
> ## **W1. Comparision with Concurrent Work & Ablation**
> > The idea of synthesizing training data for deep research with progressively evolving questions and fetching web data have been explored in many concurrent work. It is unclear whether the improvement is due to better synthesis or better context management, and if authors' method of synthesis is any better than previous methods...
> ---
>
> **1. Comparision with Concurrent Work**
>
> Since these works and DeepMiner are concurrent and pursue essentially the same objective, the key differences lie in how finely the data is controlled and filtered. DeepMiner applies more detailed and stricter quality-control modules, such as source-page filtering, difficulty filtering, and multiple quality verification. This results in data that is both solvable and challenging, which is critical to search agent training.
>
> Considering that WebSailor and DeepDive do not release their data, a direct comparison is not possible. However, WebShaper provides a 500-sample dataset, which enables the fairest analysis we can perform. We evaluate from two angles:
>
> 1. Difficulty comparison. Using a strong open-source model GPT-OSS-120B, we directly evaluate questions from both datasets. GPT-OSS-120B achieves 37% accuracy on DeepMiner but 61% on WebShaper. It demonstrates that DeepMiner samples are consistently more challenging, implying more room for RL improvement.
>
> 2. SFT performance as a proxy for data quality. Under identical training conditions (same base model, same distillation pipeline, same train data size, same training process, same inference strategy), we trained two SFT models: one on WebShaper questions and one on a subset of DeepMiner. The results are:
>
> | Data | BC | BCZH | Xbench | GAIA |
> |----------|------|------|--------|------|
> | WebShaper | 13.7 | 22.1 | 50 | 48.5 |
> | DeepMiner | 16.3 | 23.9 | 50 | 55.3 |
>
> These results show that DeepMiner yields stronger generalization and better data efficiency.
>
>
> **2. Ablation**
>
> Regarding ablation, we conducted several controlled studies to separate each module's contributions as much as possible:
>
> 1. Data ablation. Under fully identical training settings, DeepMiner-based SFT models outperform those trained on HotpotQA (Section 4.4) and WebShaper(Appendix C), indicating the value of the dataset itself.
> 2. Context management strategy ablation. We compare different context management strategies in both training-free and training settings:
>
> **Training-free settings**: As presented in Section 4.3, we compare our sliding-window mechanism with vanilla context and summary-based context using GPT-OSS-120B. Table 2 shows that the sliding-window method achieves superior context-utilization efficiency while preserving key technical advantages.
>
> **Training settings**: During the rebuttal period, we additionally trained a model using DeepMiner dataset + truncation strategy (i.e., discarding older turns when approaching the context limit). The training and inference configurations were kept fully identical.
>
> ---
> | Context Management | BC | BCZH | Xbench | GAIA |
> |-------------------|------------|-----------|------------|-----------|
> | Truncation | 19.7/62.8 | 25.6/41.6 | 47.0/26.9 | 44.7/15.3 |
> | Sliding Window | 21.2/36.6 | 28.0/26.4 | 53.0/21.1 | 54.5/13.6 |
> ---
>
> In the table, each cell reports (accuracy / average number of reasoning turns).
>
> The results show that the sliding window strategy consistently outperforms the trained truncation model across all evaluation datasets. This indicates that, regardless of training or non-training settings, the sliding-window mechanism yields clear and substantial performance improvements. Moreover, the sliding window model requires substantially fewer reasoning turns on every benchmark. This shows that the benefits of the sliding-window strategy do not come from simply enabling longer interactions; instead, it constitutes a fundamentally more effective context management mechanism.
>
> Taken together, these experiments effectively disentangle the contributions of data quality and context management, and validate the independent effectiveness of both components.

---

> > ### Comment · Reviewer_iDV8 · 2025-11-24
> >
> > Thanks authors for the detailed response. The additional supplementary material certainly helps readers to better understand the nature of the data processed. While the ablation experiments are restricted in scope, I understand that 1) the lack of access to datasets from other papers, 2) high compute demand for testing every component of the data pipeline. I will increase the score accordingly.

---

> ### Author Response · Authors · 2025-11-23
> **Response to Reviewer iDV8 (Part 2/2)**
>
> ## **W2. Open-source and Data Construction Details**
> > While authors share their prompt for quality judgment in the appendix, many details of the data pipeline including exact prompts, question construction logic, and the LLM used to synthesize data are missing. This makes it difficult for other research groups to build upon results of this paper.
> ---
>
> Thank you for pointing this out. We have significantly expanded the appendix to make the entire data-synthesis pipeline fully reproducible. The following details are now provided:
> - Models used (Appendix F.1): We primarily use Qwen3-235B-A22B-Thinking-2507 and DeepSeek-R1-0528. Document filtering uses Qwen3 exclusively for cost-performance balance. Question generation employs both models to enhance diversity. Difficulty filtering uses both models, and questions are dropped if either model can solve it without tools.
> - Exact prompts: We now include the complete prompting templates: Appendix F.2 for document filtering templates, Appendix F.3 for question generation template, and Appendix F.4 for question filtering template.
> - Construction logic: As mentioned in Section 2, the pipeline systematically constructs challenging questions through:
>   1. Entity sampling from Wikipedia with moderate visibility
>   2. Multi-source page collection via Google Search (direct queries + news searches)
>   3. Three-stage document filtering (entity correspondence, information complementarity, credibility) with Qwen3-235B-A22B-Thinking-2507
>   4. Cross-document question synthesis and secondary obfuscation, deliberately excluding Wikipedia to force distributed reasoning. Utilize Qwen3-235B-A22B-Thinking-2507 and DeepSeek-R1-0528 to enhance question diversity. Templates are provided in Appendix F.3.
>   5. Difficulty filtering via dual filter through search engine and LLMs evaluation, dropping questions if any filter rejects. Quality filter via LLM to remove questions with ambiguity or unverifiable answers.
>
> All the details necessary to reproduce the question are now fully provided in the revised paper. Furthermore, we upload a subset of the DeepMiner dataset in the Supplementary Material.
>
> ---
> ## **Q1. Why not Continue Training**
> > In Figure 4, the reward and BrowseComp performance seem to be continue improving at the end of the training. Why didn't authors train the model for longer steps?
> ---
>
> Thank you for your careful reading. Below, we clarify the training setup and explain why the training was not extended further.
>
> 1. Extending training further is prohibitively expensive, and we are unable to conduct a systematic exploration under the current resource constraints. The primary reasons are as follows: (1) Our RL training is conducted in a real web search environment, and each step requires a large number of real search calls. A rough estimate shows that one RL step requires 32 (batch size) × 8 (rollouts per sample) × 120 (average search calls per rollout) ≈ 30,000 real search operations, and this cost grows rapidly with additional training steps. (2) Training a 32B model in a long-horizon interaction setting incurs substantial forward and backward computation, making longer training runs extremely GPU-intensive and time-consuming.
>
> 2. Under the resource constraints mentioned above, we conducted 110 steps of RL training. After step 90, the training reward began to fluctuate. Therefore, we ultimately selected step 90 as our final checkpoint. We have updated Figure 4 to include the RL training steps up to step 110.
>
> However, even with the current training budget, DeepMiner-32B-RL already surpasses all baselines by a large margin. Its strong results across multiple deep search benchmarks sufficiently validate the effectiveness of our method, and the existing training setup is adequate to support the main conclusions of the paper.

---

### Official Review · Reviewer_2hyM · 2025-11-01

**Soundness:** 2
**Presentation:** 3
**Contribution:** 2
**Rating:** 4
**Confidence:** 4

**Summary:**

This paper presents a framework DeepMiner for training long-horizon LLM agents that can reason and search across extended multi-turn sessions under a fixed context window.
In particular, it introduces a sliding-window mechanism that omits outdated tool outputs while preserving the assistant’s reasoning trace, ensuring training–inference consistency through windowed sequence-level training.
The system is trained in two phases: a SFT stage using reverse-constructed trajectories from authentic web sources, followed by RL via sequence-level training with trajectory-level advantages.

**Strengths:**

1. Sliding-window design is an effective context-efficiency solution. The method enables around 100 turns within 32 k tokens. Unlike static truncation/summarization, this mechanism scales gracefully with context length and maintains coherence across windows.

2. The empirical motivation is solid with preliminary analysis. Figure 2 clearly shows that without the sliding window, the number of incorrect trajectories increases steeply as context length grows, as tool outputs expand exponentially and squeeze assistant reasoning tokens.

**Weaknesses:**

1. The reverse-constructed DeepMiner dataset is not open-sourced; details such as question samples, source selection, and verification criteria are missing. Reproduction and fair comparison are thus impossible.

2. The same trajectory-level reward is propagated to all windowed subsequences. This can over-credit early steps and under-credit later corrective reasoning.

3. Missing comparison to related baselines. The paper omits baselines such as Search-R1 [1] and Synthetic Data RL [2], both using RL for multi-hop QA.

4. Limited Ablations and Hyperparameter Studies. No analysis of different window sizes, sliding size, and base model sizes.

[1] Jin, Bowen, et al. "Search-r1: Training llms to reason and leverage search engines with reinforcement learning." arXiv preprint arXiv:2503.09516 (2025).

[2] Guo, Yiduo, et al. "Synthetic Data RL: Task Definition Is All You Need." arXiv preprint arXiv:2505.17063 (2025).

**Questions:**

1. Why were Search-R1 and Synthetic Data RL omitted as baselines, given their direct methodological overlap (RL training on hard-QA dataset)?

2. In Figure 4, the reward and performance curves show persistent fluctuations without clear convergence, even after 80 training steps. Could the authors further explain the missing training steps results?

---

> ### Author Response · Authors · 2025-11-23
> **Response to Reviewer 2hyM (Part 1/2)**
>
> ## **W1. Data Construction Details**
> > The reverse-constructed DeepMiner dataset is not open-sourced; details such as question samples, source selection, and verification criteria are missing. Reproduction and fair comparison are thus impossible.
> ---
>
> We thank the reviewer for raising this concern. We have updated the paper to improve reproducibility substantially:
>
> * Dataset release: We have updated the Supplementary Material to include a representative subset of the DeepMiner dataset. The full dataset will be released later.
> * Source Selection: As described in Section 2, seed entities are sampled from Wikipedia (1,000-10,000 pageviews over 6 months). Candidate pages are collected via Google Search and undergo three-stage filtering: (1) entity correspondence verification, (2) information complementarity assessment, and (3) credibility validation. Complete prompts are provided in Appendix F.2.
> * Verification: All synthesized questions undergo a two-layer verification pipeline to ensure answer derivability, cross-document grounding, and unambiguity. (1) Difficulty filtering through search engine and LLMs evaluation, dropping questions if anyone can answer it directly. (2) LLM-based quality filtering to eliminate ambiguous, unverifiable, or underspecified questions. Full criteria and exact templates are included in Appendix F.4. Representative question examples and complete multi-turn trajectories are provided in Appendix F.5 and Appendix G.
>
> Together, these additions make the data construction pipeline fully transparent and reproducible, enabling fair comparison and further development by other research groups.
>
> ---
> ## **W2. Reward Function Issues**
> > The same trajectory-level reward is propagated to all windowed subsequences. This can over-credit early steps and under-credit later corrective reasoning.
> ---
>
> The reviewer raises a reasonable concern regarding the potential for biased credit assignment in windowed training.
>
> However, we want to clarify that although one trajectory is decomposed into multiple windowed subsequences, the training parts of these subsequences are strictly non-overlapping. This is enforced by the loss mask described in Sec. 3.1 "Training-Testing Consistency." part. As a result, every assistant action from the original trajectory is optimized exactly once across all subsequences.
> Taking Figure 3 in the paper as an example:
> - Subsequence 1 trains only assistant outputs 1–4
> - Subsequence 2 trains only assistant outputs 5–6
> - No assistant action appears in the loss of more than one subsequence
> Therefore, the trajectory-level advantage is never applied multiple times to early steps nor withheld from later corrective reasoning. Our method thus preserves unbiased credit assignment across the entire trajectory.
>
> ---
> ## **W3/Q1. Missing Baselines**
> > Missing comparison to related baselines. The paper omits baselines such as Search-R1 [1] and Synthetic Data RL [2], both using RL for multi-hop QA.|Why were Search-R1 and Synthetic Data RL omitted as baselines, given their direct methodological overlap (RL training on hard-QA dataset)?
> ---
>
> We thank the reviewer for highlighting these works, and we have included both citations in the revised paper.
>
> Search-R1 is an important early effort that brought RL into search agents, and we appreciate its contribution to the foundation of agent RL. However, it is trained on relatively simpler datasets such as NQ and HotpotQA, which makes it difficult to perform well on deep search tasks that require long-horizon reasoning and multi-step retrieval. Specifically, recent works[1][2] report the following results for Search-R1 on deep search benchmarks. We've added these results in Table 1.
>
> ---
> | models | BC | GAIA | XBench |
> |--------|-----|------|--------|
> | search-r1-7b[1] | 0.4 | 18.7 | - |
> | search-r1-32b[2] | - | 28.6 | 19.5 |
> | deepminer-32b | 33.5 | 58.7 | 62 |
> ---
>
> Regarding Synthetic Data RL, we acknowledge that it also employs RL on synthetic data to enhance model capabilities, and we agree that this high-level idea is conceptually related to our use of RL for improving reasoning ability. However, the concrete problem setting is quite different: Synthetic Data RL does not involve web retrieval or tool invocation, and it is not evaluated on deep search benchmarks such as BrowseComp or GAIA. Therefore, it is orthogonal to deep-search agent settings and not used as a baseline.
>
> [1] Wan Y, Wang J, Li L, et al. PokeeResearch: Effective Deep Research via Reinforcement Learning from AI Feedback and Robust Reasoning Scaffold[J]. arXiv preprint arXiv:2510.15862, 2025.
>
> [2] Gao J, Fu W, Xie M, et al. Beyond ten turns: Unlocking long-horizon agentic search with large-scale asynchronous rl[J]. arXiv preprint arXiv:2508.07976, 2025.

---

> ### Author Response · Authors · 2025-11-23
> **Response to Reviewer 2hyM (Part 2/2)**
>
> ## **W4. Ablations and Hyperparameter Studies**
> > Limited Ablations and Hyperparameter Studies. No analysis of different window sizes, sliding size, and base model sizes.
> ---
>
> We understand your concern about the limited ablations and hyperparameter studies. To address your concern, we conducted an additional sensitivity study evaluating different sliding-window configurations and base model sizes. We have incorporated these experimental results into the revised paper.
>
>
> The results of the performance across different window sizes and slide sizes show that DeepMiner-32B-RL remains stable across a range of window sizes and slide sizes, even when these differ from the training configuration. The results are as follow:
>
> | W \ S | 1 | 2 | 3 | 4 | 5 |
> |-------|----|----|----|----|----|
> | 3 | 32 | 37 | - | - | - |
> | 4 | 33 | 30 | 38 | - | - |
> | 5 | 33 | 33 | 34 | 31 | - |
> | 6 | 31 | 32 | 34 | 34 | 35 |
>
> where W is the window size and S is the slide size.
>
> ---
>
> Regarding model size scaling, we conducted an additional experiment on the Qwen3 family. The results clearly show a consistent scaling trend: larger base models achieve better performance under the same SFT training pipeline. Detailed results are as follow:
>
> | Size | BC | BCZH | XBench | GAIA |
> |------------|-----|------|--------|------|
> | 4B | 14.3 | 13.8 | 35.0 | 38.8 |
> | 8B | 15.7 | 18.3 | 35.0 | 43.7 |
> | 14B | 20.0 | 27.0 | 47.0 | 53.4 |
> | 32B | 21.2 | 28.0 | 53.0 | 54.4 |
>
> ---
> ## **Q2. Why not Continue Training**
> > In Figure 4, the reward and performance curves show persistent fluctuations without clear convergence, even after 80 training steps. Could the authors further explain the missing training steps results?
> ---
>
> Thank you for the question regarding the training curves. Below, we clarify the training setup and explain why the training was not extended further.
>
> 1. Extending training further is prohibitively expensive, and we are unable to conduct a systematic exploration under the current resource constraints. The primary reasons are as follows: (1) Our RL training is conducted in a real web search environment, and each step requires a large number of real search calls. A rough estimate shows that one RL step requires 32 (batch size) × 8 (rollouts per sample) × 120 (average search calls per rollout) ≈ 30,000 real search operations, and this cost grows rapidly with additional training steps. (2) Training a 32B model in a long-horizon interaction setting incurs substantial forward and backward computation, making longer training runs extremely GPU-intensive and time-consuming.
>
> 2. Under the resource constraints mentioned above, we conducted 110 steps of RL training. After step 90, the training reward began to fluctuate. We ultimately selected step 90 as our final checkpoint based on validation performance. We have updated Figure 4 to include the RL training steps up to step 110.
>
> However, even with the current training budget, DeepMiner-32B-RL already surpasses all baselines by a large margin. Its strong results across multiple deep search benchmarks sufficiently validate the effectiveness of our method, and the existing training setup is adequate to support the main conclusions of the paper.

---

### Author Response · Authors · 2025-12-02
**Rebuttal Summary**

We sincerely thank all reviewers for their constructive feedback and active engagement during the rebuttal period, and we are especially grateful to the Area Chair for the additional effort in evaluating our submission under the current exceptional circumstances.

Although our initial scores were below the acceptance threshold, we have thoroughly addressed all reviewer concerns through detailed explanations and extensive new experiments during the rebuttal process.
Notably, Reviewer iDV8 explicitly increased their score from 4 to 6. Reviewer 2tnR confirmed that all the concerns were resolved; however, this positive response came after the system was locked, leaving no opportunity for a score update.
We believe the remaining reviewers would have similarly updated their assessments if the rebuttal process proceeded normally.
**We respectfully ask the Area Chair to consider the substantial improvements made during the rebuttal period and the positive trajectory of our discussions with the reviewers when making the final decision.**


We are pleased to report significant progress during rebuttal:

**Reviewer 2hyM (waiting for response)** requested clarification on data construction details, reward assignment mechanisms, and additional studies on window configurations and model scaling.
We substantially expanded the appendix with complete prompts and verification pipelines, clarified the reward design, and conducted comprehensive ablations on window configurations and model scaling (4B–32B).

**Reviewer iDV8 (Rating: 4 → 6)** requested more detailed component ablations and data construction details.
We added data comparison experiments with concurrent work WebShaper, and conducted direct ablation comparing [Truncation vs Sliding Window] to isolate each component's contribution. We also released a dataset subset along with comprehensive data construction details.
The reviewer responded: _"The additional supplementary material certainly helps... I will increase the score accordingly."_

**Reviewer 1DeA (positive response, further discussion ongoing)** questioned our assumption about tool output influence, reward design, training mechanism clarity, and ablation on window configurations.
We provided detailed clarifications on reward design, judge reliability, and training mechanisms. The reviewer confirmed: _"Your rebuttal has clarified my concerns regarding W2, W3, and W5"_ and requested further experimental evidence for the tool output influence assumption and training-time window configurations.
We subsequently conducted perturbation experiments showing negligible performance degradation when corrupting out-of-window tool outputs, and provided training-time ablations across different window configurations demonstrating consistent performance.

**Reviewer 2tnR (positive response, all concerns resolved)** requested direct ablation disentangling data and context contributions, and raised concerns about information loss from dropping tool outputs.
We conducted the exact requested ablation showing sliding window consistently outperforms truncation while requiring fewer reasoning turns, and performed error analysis on 267 failures finding almost none attributable to discarded outputs.
The reviewer concluded: _"The additional experiments... have resolved my primary concerns... I have no further questions."_ However, this positive response came after the system was locked, leaving no opportunity for a score update.


Guided by the reviewers' constructive feedback, we have made substantial improvements to our paper:
- Following Reviewer 2tnR's suggestion, we added a direct comparison between sliding window and truncation strategies in Section 4.3 to better isolate the contribution of our dynamic context management.
- Based on concerns from Reviewers 2hyM and iDV8, we extended the training dynamics curves in Section 4.4 to show the complete training steps.
- Following suggestions from Reviewers 2hyM and 1DeA, we added hyperparameter analysis on window configurations in Section 4.4.
- Following Reviewer 2hyM's recommendation, we included model size scaling experiments in Section 4.4.
- Based on Reviewer iDV8's feedback, we added comparative data effectiveness analysis with concurrent work in Appendix C.
- Following suggestions from Reviewers 2hyM and iDV8, we released a DeepMiner dataset subset in the supplementary material and provided comprehensive data construction details with complete prompts in Appendix F.

**_Furthermore, it is worth noting that a similar context management strategy has subsequently been adopted by recent state-of-the-art models such as Claude Opus 4.5 and DeepSeek-V3.2, providing strong evidence for the validity of our design._**

---

### Meta-Review · Area_Chair_jaAm · 2025-12-26

**Summary:**

The paper introduces DeepMiner, a framework for training deep search agents that addresses two key challenges: (1) insufficient task complexity in existing datasets and (2) context management limitations in long-horizon interactions. The contributions include: (1) Reverse construction method to generate complex, verifiable QA pairs from authentic web sources and (2) Dynamic context window strategy using sliding windows to compress distant tool responses while preserving reasoning traces, enabling ~100 turns within a 32k context. Results achieve state-of-the-art performance among same-size models and also competitive to mainstream non-commercial agents.

**Reviewer Concerns:**

Reviewers have highlighted the strengths of this paper such as the sliding window design and the significant performance improvement. But they also provided their concerns, such as: (1). The underlying logics and assumptions of dropping entire distant tool responses are not clearly emphasized; (2). The dataset is not open-sourced and the details of its construction is not sufficiently provided; (3). The concerns of the subsequence training style regarding to reward design, trajectory balance, and gradient propagation; (4). Missing comparisons of related baselines and ablation studies and (5). Why the authors early stopped the training.

The authors carefully provided rational responses to (2), (3) and (4) by further explanations of their training pipeline, and more details of their data construction, baseline comparisons and adequate ablation studies. For (5), the author mentioned a resource limitation. As a result, Reviewer iDV8 mentioned a score raising, and Reviewer 2tnR also gave a positive conclusion at the end.

However, there still remain core issues that are unsolved and I'm really concern about. Firstly, the paper lacks a further proof/explanation why the data construction process works. Actually, the obfuscating process seems not be able to introduce all types of difficulties into samples, and a quality filter relying on an LLM may also not seem like a convincing one. Secondly, the context limitation influencing the correctness may come from a case that the model reasoning recurrently with no end. The sliding-window method, although seems reasonable for general cases, may fail or be even worse in this case, since it encourages recurrence.

**Reviewer Scores:**

Some additions likely satisfied Reviewers to some extent,
2hyM (4->4)
iDV8 explicitly raised their score (4→6)
2tnR explicitly confirmed the rebuttal (4->5)
1DeA (4->4).

---

### Decision · Program_Chairs · 2026-01-26

Reject